# Adipose stem cell niche reprograms the colorectal cancer stem cell metastatic machinery

Simone Di Franco [1], Paola Bianca [2], Davide Stefano Sardina [2], Alice Turdo [2], Miriam Gaggianesi [1], Veronica Veschi [1], Annalisa Nicotra [2], Laura Rosa Mangiapane [2], Melania Lo Iacono [1], Irene Pillitteri [1], Sander van Hooff[3], Federica Martorana [4], Gianmarco Motta [4], Eliana Gulotta[5], Vincenzo Luca Lentini [6], Emanuele Martorana [7], Micol Eleonora Fiori [8], Salvatore Vieni[1], Maria Rita Bongiorno[2], Giorgio Giannone[9], Dario Giuffrida[9], Lorenzo Memeo [9], Lorenzo Colarossi [9], Marzia Mare[9,10], Paolo Vigneri [11], Matilde Todaro[2], Ruggero De Maria[12,13], Jan Paul Medema[3,14] & Giorgio Stassi [1✉]

Obesity is a strong risk factor for cancer progression, posing obesity-related cancer as one of the leading causes of death. Nevertheless, the molecular mechanisms that endow cancer cells with metastatic properties in patients affected by obesity remain unexplored.

Here, we show that IL-6 and HGF, secreted by tumor neighboring visceral adipose stromal cells (V-ASCs), expand the metastatic colorectal (CR) cancer cell compartment (CD44v6 + ), which in turn secretes neurotrophins such as NGF and NT-3, and recruits adipose stem cells within tumor mass. Visceral adipose-derived factors promote vasculogenesis and the onset of metastatic dissemination by activation of STAT3, which inhibits miR-200a and enhances ZEB2 expression, effectively reprogramming CRC cells into a highly metastatic phenotype. Notably, obesity-associated tumor microenvironment provokes a transition in the transcriptomic expression profile of cells derived from the epithelial consensus molecular subtype (CMS2) CRC patients towards a mesenchymal subtype (CMS4). STAT3 pathway inhibition reduces ZEB2 expression and abrogates the metastatic growth sustained by adipose-released proteins. Together, our data suggest that targeting adipose factors in colorectal cancer patients with obesity may represent a therapeutic strategy for preventing metastatic disease.

[1] Department of Surgical Oncological and Stomatological Sciences (DICHIRONS), University of Palermo, Palermo, Italy. [2] Department of Health Promotion Sciences, Internal Medicine and Medical Specialties (PROMISE), University of Palermo, Palermo, Italy. [3] Laboratory for Experimental Oncology and Radiobiology, Center for Experimental and Molecular Medicine, Cancer Center Amsterdam, Amsterdam UMC, University of Amsterdam, Amsterdam, The Netherlands. [4] Department of Clinical and Experimental Medicine, A.O.U. Policlinico-Vittorio Emanuele, Division of Medical Oncology, University of Catania, Catania, Italy. [5] Reconstructive Plastic Surgery, ARNAS Ospedali Civico Di Cristina e Benfratelli, Palermo, Italy. [6] Pathology Unit, Ospedali Riuniti Villa Sofia-Cervello, Palermo, Italy. [7] IOM (Mediterranean Institute of Oncology) Ricerca, Viagrande, Catania, Italy. [8] Department of Oncology and Molecular Medicine, Istituto Superiore di Sanità, Rome, Italy. [9] Department of Experimental Oncology, Mediterranean Institute of Oncology, Viagrande, Catania, Italy. [10] Department of Clinical and Experimental Medicine, University of Messina, "G.Martino" Hospital, Messina, Italy. [11] Department of Clinical and Experimental Medicine, A.O.U. Policlinico-Vittorio Emanuele, Center of Experimental Oncology and Hematology, University of Catania, Catania, Italy. [12] Istituto di Patologia Generale, Università Cattolica del Sacro Cuore, Rome, Italy. [13] Fondazione Policlinico Universitario "A. Gemelli" - I.R.C.C.S., Rome, Italy. [14] Oncode Institute, Amsterdam UMC, University of Amsterdam, Amsterdam, The Netherlands. ✉email: giorgio.stassi@unipa.it

Colorectal cancer (CRC) is the third most common cause of cancer-related death worldwide[1]. Despite the great effort made in the study of CRC, the molecular mechanisms underlying the metastatic process are still poorly defined. It is becoming increasingly clear that obesity, whose prevalence is raising worldwide[2], is associated with cancer incidence and contributes to up to 20% of cancer-related deaths[3,4]. Adipose tissue (AT) is an endocrine organ subdivided into two compartments. White AT, prevalently distributed subcutaneously and surrounding bowels, and brown AT, present in the cervical and supraclavicular area, are endowed with fat storage and thermogenic function, respectively[5]. White AT is characterized by a marked cell variety, which includes adipocytes, immune cells, vascular, and progenitor cells[6]. In obesity state, both the subcutaneous and visceral white adipose tissues expand by hypertrophy of pre-existing adipocytes. Recent evidence showed that visceral fat, in high-fat diet-induced obese mouse models, determines a hyperplastic response, which is driven by adipose precursor cells, identified as Lin$^-$/Sca1$^+$/CD29 + /CD34 + [7]. This phenomenon could be due to the different embryological origin of subcutaneous and visceral adipose tissue, and/or by the presence of specific resident factors, as highlighted by lineage tracing experiment performed in adult Wt1-GFP knock-in mice[8]. In particular, mature adipocytes are postmitotic cells, suggesting that hyperplasia arises from the expansion and differentiation of adipocyte precursors[9,10]. Mature adipocytes together with adipose stromal cells (ASCs) influence the surrounding cell populations through the release of a plethora of inflammatory and angiogenic cytokines[5,7,11,12]. In individuals affected by obesity, adipose-released proteins, including TNF-α, IL-6, and monocyte chemoattractant protein1 (MCP1), promote a chronic inflammatory state that creates a microenvironment able to sustain tumor progression[13,14]. In the presence of obesity-released protein, various types of cancers activate cell proliferative processes and behave more aggressively[15–22]. Several retrospective studies analyzing large cohorts of cancer patients highlighted that obesity has a significant impact on overall survival, positing this indicator as a significant negative prognostic factor, including for CRC patients[23–25]. Furthermore, Bhaskaran and colleagues showed that while in CRC patients, characterized by a BMI from 15 to 25 kg/m$^2$, the mortality risk does not vary, it linearly increases in those with BMI from 25 to 50 kg/m$^{2,25}$. An increase of VAT governs the expansion of intestinal stem cells and proliferation of progenitors by activating GSK-3β and contributing to β-catenin accumulation in canonical WNT signaling pathway[26,27]. It has been established that cancers, including colon, originate by a small subset of cells with stem-like features called cancer stem cells (CSCs), whose phenotype and behavior can be modulated by the tumor microenvironment (TME)[28–30]. More recently, TME cytokines, such as IL-6 and HGF, were demonstrated to have clinical relevance and induce cancer cell stemness concomitantly enhancing the epithelial-to-mesenchymal transition (EMT), cell migration, and metastatic potential[31–33]. The interaction between the CRC cells and their TME is fundamental for addressing cancer cell fate, which characterizes the tumor biological behavior[34]. In agreement, CRC can be stratified into four consensus molecular subtypes (CMS) according to their molecular signature, which in part depends on the composition of the TME and predicts clinical outcome[35]. Specifically, CMS1 displays an immune system signature, CMS2 represents an epithelial-like cancer characterized by the activation of Wnt and c-Myc signaling pathways, CMS3 shows a metabolic deregulation, and CMS4 displays a mesenchymal phenotype. Of note, 14% of all CRCs represents a transition or mixed phenotype. The dynamic CRC microenvironment that induces a CMS plasticity shaping the clinical outcome, has been poorly investigated.

In this work, we have investigated the paracrine effect of VAT in the mesenchymal phenotype modulation of CRC cells, by reprogramming the CMS. We show that in CRC patients affected by obesity, tumor-infiltrating ASCs are key elements of cancer progression. CRC cell-released NGF and VEGF induce ASC recruitment and transdifferentiation in endothelial cell phenotype. Moreover, we provided evidence that IL-6 and HGF, enhance the tumorigenic and metastatic potential of CRC cells. Our data indicate microenvironmental cytokines as essential prognostic molecules that predict the tumor behavior of obesity-associated cancer patients.

## Results

**Visceral adipose stromal cells promote tumorigenic and metastatic activity of CR-CSphCs.** To confirm the clinical impact of obesity on the biological behavior of cancer cells, we first evaluated the potential role of adipose tissue in CRC progression, correlating recurrence and survival in CRC patients with a healthy weight (18.50 < BMI < 25) or affected by obesity (BMI > 30). A meta-analysis of a large cohort of CRC patients revealed that obesity is negatively associated with survival probability (Fig. 1a). Importantly, this correlation was also confirmed by the progression-free survival (PFS) analysis on a large cohort of CRC patients, positing BMI as a negative prognostic factor independent of stage and treatment (Fig. 1b, and Supplementary Fig. 1a, b). Obesity is characterized by a chronic low-grade inflammation, which relies on the presence of an heterogenous cell population, including lymphocytes, endothelial cells, macrophages, progenitor, and mature adipose cells[36]. Intriguingly, immunohistochemical analysis of tumor sections from CRC patients with obesity highlighted that adipose cells, marked by high expression of adiponectin, were located at tumor invasive front and interspersed among tumor cells expressing CDX2 and covering the 28% of the entire tumor mass (Fig. 1c and Supplementary Fig. 1c, d). Likewise, CRC liver metastasis of patients affected by obesity displayed a prominent presence of adipose cells at metastatic lesion edge, in the proximity of peritumoral budding. This phenomenon was not observable in primary and metastatic CRC tissues from lean patients (Fig. 1c and Supplementary Fig. 1c, d). As the obesity inflammatory environment is strongly sustained by the paracrine activity of mature adipocytes and adipose-derived vascular stromal cells[37], including adipose stem cells, we next investigated whether this cell subset is present in the tumor area and could influence cancer cell phenotype. By immunohistochemical analysis, we found that a high percentage of cells within tumor-neighboring and -infiltrating adipose tissue of patients with obesity are CD34 + /CD45$^-$, while lacking the expression of CD31, indicating their ASC identity in both primary and metastatic CRC. Tumor specimens of lean CRC patients displayed the presence of cells expressing CD34 + / CD31 + /CD45$^-$ ascribable to an endothelial phenotype (Fig. 1d and Supplementary Fig. 1e).

Whereas ASCs isolated from subcutaneous AT (S-ASCs) are enriched in CD10 positive cells, visceral ASCs (V-ASCs) show high expression of CD200 and WT1[8,38] (Supplementary Fig. 1f, g and Supplementary Table 1). Following exposure to adipogenic differentiation medium, S-ASCs and V-ASCs were equally inclined to differentiate in vitro toward adipocytes, as displayed by AdipoRed staining (Supplementary Fig. 1h, i)[39,40]. Conditioned medium (CM) from V-ASCs boosted both the colony-forming capacity of CRC sphere cells (CR-CSphCs) and the in vivo tumor growth, which was sustained by the high number of cells expressing Ki67 observed in tumor xenografts generated by the co-injection of CR-CSphCs and V-ASCs (Supplementary Fig. 1j–l and Supplementary Table 2). We next investigated

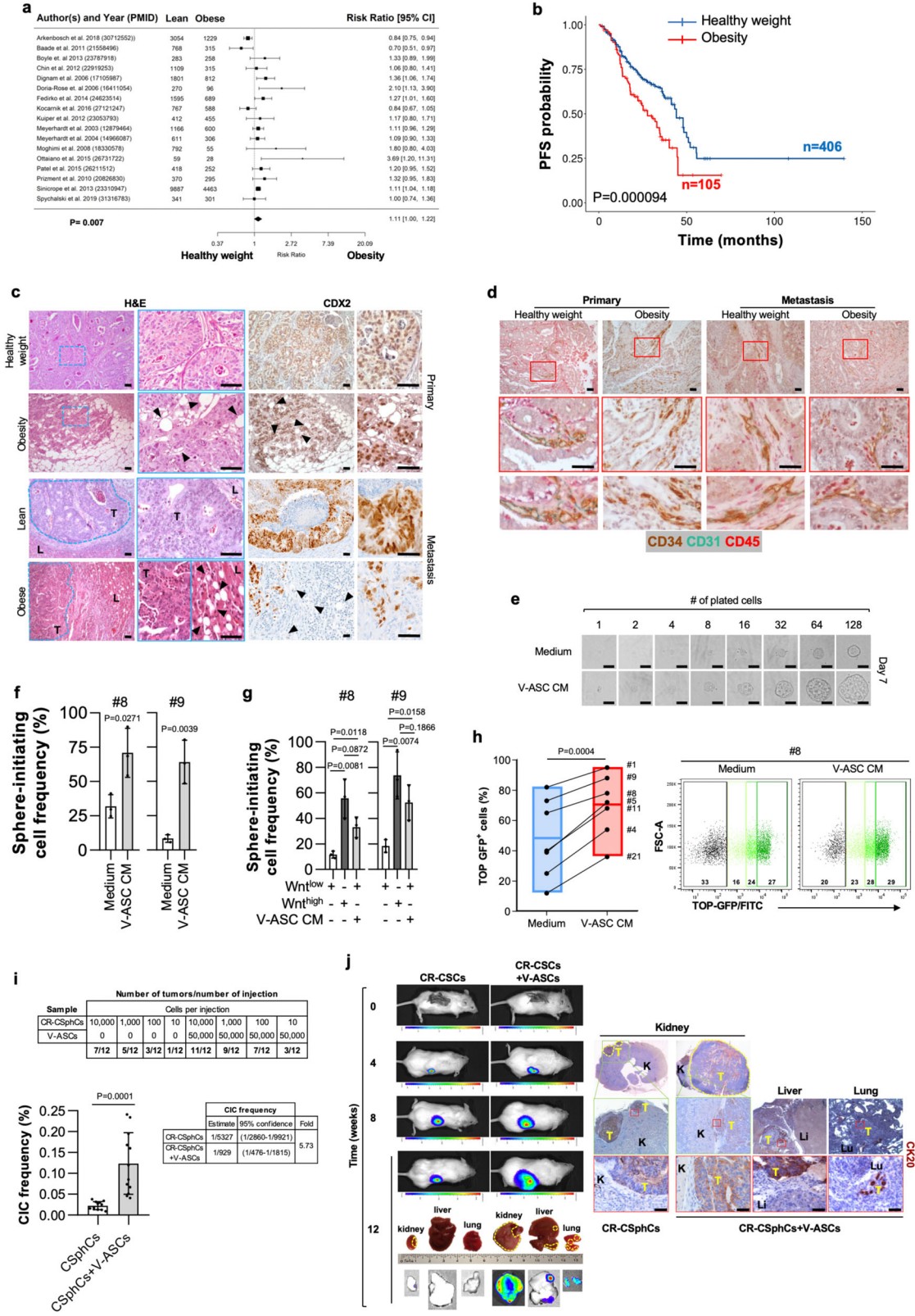

whether the adipogenic niche could impact the self-renewal activity of CRC cells. The in vitro limiting-dilution assay showed that treatment with V-ASC-released factors enhances the sphere-initiating cell frequency of bulk and enriched Wnt$^{low}$ CR-CSphCs, recapitulating the clonogenic potential of Wnt$^{high}$ cells (Fig. 1e–g). This phenomenon was paralleled by the conversion of Wnt$^{inactive}$ (GFP$^-$) to Wnt$^{active}$ (GFP +) cells, confirming the

crucial role of the Wnt signaling pathway in the CRC stemness-associated reprogramming (Fig. 1h).

To explore the role of ASCs more in detail, we investigated their effect in the clonal tumor initiation of CR-CSphCs. V-ASCs co-injected into the subrenal capsule of mice with a limiting-dilution series of CR-CSphCs increased the frequency of cancer-initiating cells (5.73-fold; $p < 0.0001$) (Fig. 1i). These data indicate

**Fig. 1 Tumor-infiltrating VAT boosts the metastatic potential of CR-CSphCs. a** Forest plot of survival changes in high (>30, obesity) versus low (≥18.5 and <25, lean) BMI CRC patients. Data represent the risk ratio ± 95% CI. Statistical significance was calculated by a Random-effect meta-analysis model. **b** Kaplan–Meier of progression-free survival (PFS) curve in a cohort of 511 CRC patients, based on BMI status. Healthy weight indicates 18,5<BMI < 30, and obesity BMI > 30. Statistical significance was calculated using the log-rank (Mantel–Cox) test. **c** H&E analysis and CDX2 expression on primary and liver metastasis in CRC patients with healthy weight or affected by obesity. Black arrow heads indicate tumor-infiltrating adipose cells. Li: liver; T = tumor. **d** Immunohistochemical analysis of CD34 (brown color), CD31 (green color), and CD45 (red color) in tissues as in **c**. For **c**, **d** one representative of 9 independent experiments is shown. **e** Phase-contrast analysis of CMS2 cells (CSphC #9) treated with medium or V-ASC CM. For (**c–e**) scale bars, 100 μm. One representative of three independent experiments is shown. **f** ELDA software analysis of the clonogenic activity in CMS2 CR-CSphCs following treatment with medium or V-ASC CM. **g** Clonogenic assay of CMS2 CR-CSphC lines TOP–GFP^high and TOP–GFP^low (15% highest/lowest TOP–GFP levels) treated with medium or V-ASC CM. For (**f–g**) statistical significance was calculated using the two-tailed t test and data are mean ± standard error of three independent experiments performed with CR-CSphCs isolated from three different CRC patients (CSphC #8, #9). **h** Percentage of TOP–GFP positive cells, in CMS2 cells treated with medium or V-ASC CM (left panel). Box plots show min-to-max values, with line indicating the mean value. Flow cytometry analysis of TOP–GFP (black color indicates Wnt⁻ cells; green color scale indicates low, intermediate, and high Wnt⁺ cells) (right panel). Statistical significance was calculated using the paired two-tailed t test. Data are mean ± standard error of independent experiments performed with different CR-CSphCs (#1, #4, #5, #8, #9, #11, #21). **i** Number of mouse tumor xenografts generated by subrenal capsule injection of 10, 100, 1000, or 10,000 CR-CSphCs, alone or in combination with 50,000 V-ASCs (upper panel). Percentage of cancer-initiating cell (CIC) and its fold increase of cells (lower panels). Data are mean ± standard error (95% confidence interval) of 12 independent experiments performed with CR-CSphCs injected as described above. Statistical significance was calculated by ELDA software (http://bioinf.wehi.edu.au/software/elda/). **j** In vivo imaging and CK20 immunohistochemistry analysis of xenograft tumor formation obtained by subrenal capsule injection of 100 CR-CSphCs alone or together with V-ASCs at the indicted time points. Photon signal of all metastatic sites (kidney, liver, and lungs) at 12 weeks. A yellow dotted line indicates a tumor xenograt lesion. Tumor (T), kidney (K), liver (Li), and lung (Lu) are indicated. One representative of 12 independent experiments is shown. Scale bars, 100 μm.

that CRC spheres retain cells endowed with stem cell properties, which are significantly expanded in presence of adipose microenvironment stimuli.

Interestingly, V-ASC is endowed with migration capacity (Supplementary Fig. 1n, o). Their CM significantly enhanced in vitro cell invasion of CR-CSphCs (Supplementary Fig. 1m). In line with the more pronounced invasive potential, in vivo limiting-dilution assay, in presence of V-ASCs, showed that CMS2 CR-CSphCs acquire the ability to generate metastatic lesions into the liver and lungs even when a small number of cells is transplanted (Fig. 1j). While S-ASCs were unable to influence the liver and lung colonization, V-ASCs potentiate the migratory capacity and the engraftment of CR-CSphCs in distant sites (Supplementary Fig. 1p, q). These data indicate a crosstalk between ASCs and CRC cells in supporting cancer progression.

**IL-6 and HGF expand the number of CD44v6 + CR-CSCs, which produce NGF and favor the migration capacity and endothelial transdifferentiation of visceral ASCs.** To identify the key players of metastasis promoted by VAT, we next investigated whether cytokines released by V-ASCs would influence the metastatic properties of CR-CSphCs. According with reported literature, V-ASCs, isolated from CRC patients with obesity, produce more abundant IL-6 and HGF as compared with S-ASCs, CR-CSphCs, and primary adipose tissue cells (AT) (Freese et al., 2015) (Fig. 2a). CR-CSphCs and their differentiated counterpart (sphere-derived adherent cells, SDACs), obtained as previously described[41] and characterized respectively by high and low expression of active β-catenin[42] (Supplementary Fig. 2a), constitutively expressed IL-6R and c-MET (Supplementary Fig. 2b, c). The presence of IL-6 and/or HGF enhanced the proliferative capacity and colony-forming potential of CR-CSphCs (Fig. 2b, c), which concomitantly acquire an invasive phenotype together with the expression of stemness and metastasis-related genes such as *WNT5A*, *WNT5B*, *WNT7A*, *MMP2*, *MMP9*, *TWIST*, *NODAL*, *SDF1* and *ZEB2* (Fig. 2d, e and Supplementary Fig. 2d). IL-6 and HGF increased ZEB2 protein expression, in line with the upregulation of its mRNA levels (Fig. 2f). Moreover, blockade of IL-6 and/or HGF abrogated cell proliferation and migratory activity of CR-CSphCs induced by V-ASCs-released proteins (Fig. 2g, h). Accordingly, V-ASC-derived IL-6 and HGF released into V-ASC CM significantly boosted the

expression of CD44v6 (Fig. 2i), a marker that identifies CRC cells characterized by a robust metastatic potential[43], and even induced CD44v6⁻ sphere cells to acquire CD44v6 expression (Fig. 2j and Supplementary Fig. 2e).

Because ASCs are characterized by the expression of CD271[44–46], we evaluated the involvement of its ligands, produced by CSCs, in the recruitment of ASCs. CD44v6 + cells express high levels of nerve growth factor (NGF) and NTF3 mRNA as compared with the other CD271 ligand family members, BDNF and NTF4 (Fig. 2k) and secrete NGF and NT-3, while these neurotrophins are barely released by CD44v6⁻ cells, unless exposed to CM of V-ASC (Fig. 2l). This phenomenon is likely due to the enhancement of neurotrophins production and the reprogramming of CD44v6⁻ into CD44v6 + cells, mediated by HGF/c-MET signaling pathway[43,47]. The CD44v6 + cell-released NGF and NT-3 promote ASCs recruitment, which is completely prevented by a CD271 neutralizing antibody, compared with the weaker effects of NGF neutralizing antibody (Fig. 2m). Moreover, CR-CSphCs exposed to V-ASC CM were also able to attract ASCs (Fig. 2m, n). Notably, the analysis of a cohort of 289 CRC patients showed a significant negative correlation between relapse-free survival (RFS) and CD271 expression (Supplementary Fig. 2f, g). Thus, these data suggest that ASC-derived IL-6 and HGF can reprogram CD44v6⁻ progenitors into CD44v6 + cells, which can increase their metastatic potential by secreting NGF and recruiting ASCs within tumor mass.

Because CD44v6 + cells express and produce high levels of VEGF, which augments the proliferation rate of ASCs (Fig. 3a, b and Supplementary Fig. 3a), we hypothesized that its release could also promote angiogenesis and vasculogenesis. A global RNA-Seq transcriptomic analysis and functional enrichment of differentially expressed genes (DEGs) computed by Panther of CD44v6 + versus CD44v6⁻ cells highlighted enrichment in biological processes associated with extracellular matrix (ECM) remodeling (Supplementary Fig. 3b). The majority of CD271 + ASCs express VEGFR (Fig. 3c), suggesting that CD44v6 + cell-derived VEGF may trigger an angiogenic signal on ASCs. The paracrine activity of CD44v6 + cells induced the transdifferentiation of enriched CD34 + /CD31⁻ASCs toward endothelial-like cells expressing CD31 (Fig. 3d and Supplementary Fig. 3c). Moreover, exposure to CM released by CD44v6 + cells led HUVEC endothelial cells to develop vascular

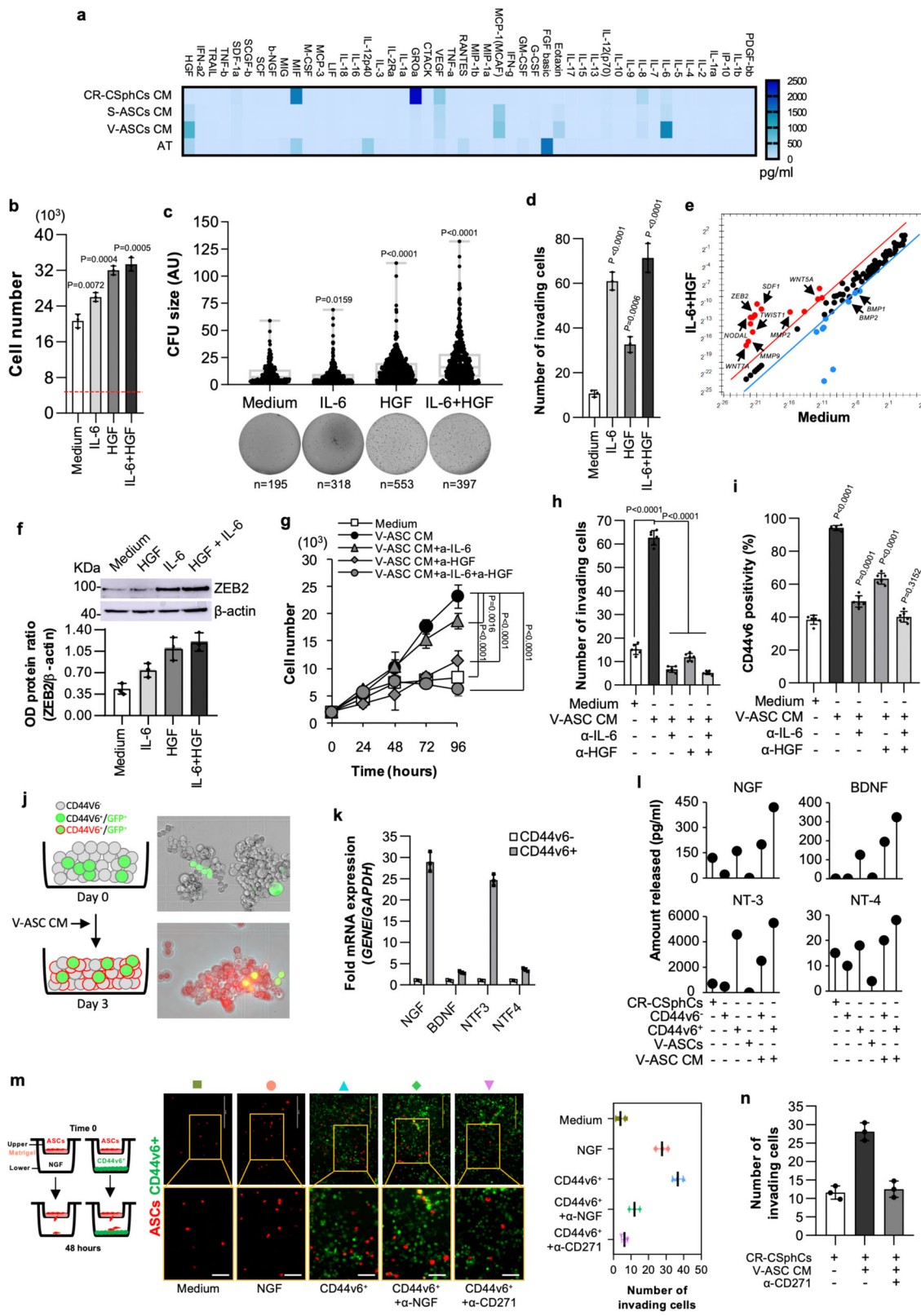

tubules likewise those formed following treatment with VEGF (Fig. 3e and Supplementary Fig. 3d). In line with the more pronounced vascular density, associated with a preferential localization of CD44v6 + cells at the vascular front and observed in CRC xenografts generated by co-injection of CR-CSphCs and V-ASCs (Fig. 3f), these findings suggest the involvement of the CSC compartment in the neo-angiogenesis and angiogenic sprouting of pre-existing capillaries.

**Visceral released proteins induce EMT of CR-CSphCs.** Next, we sought to investigate whether the presence of ASCs within tumor mass could lead to the acquisition of a transcriptional signature

**Fig. 2 Adipose-derived factors expand CD44v6 + cell fraction that secretes NGF and potentiates the migration capacity of ASCs. a** Cytokines secreted by CR-CSphCs (n = 4: #1, #8, #9, #21), S-ASCs (n = 6: #3, #5, #6, #8, #14, #20), V-ASCs (n = 6: #3, #5, #6, #8, #14, #18), or primary adipose tissue (AT) (n = 4). Data are the mean of 3 independent experiments. **b** Cell growth of CR-CSphCs treated for 5 days with IL-6 and HGF alone or in combination. The dotted red line shows the cell number at day 0. **c** Colony size of CR-CSphCs treated as indicated. n represents the number of colonies. Statistical significance was calculated using the two-tailed t test. **d** Invasion assay of CR-CSphCs pretreated with the indicated cytokines for 48 h. For **b–d** data show mean ± S.D. of three independent experiments using fourdifferent CR-CSphCs (CSphC #1, #8, #9, #21). **e** mRNA expression levels of CSC-related genes in CMS2 CR-CSphCs (CSphC #8, 9) exposed to vehicle (Medium) or IL-6 in combination with HGF for 48 h. **f** Immunoblot analysis of ZEB2 in CMS2 CR-CSphCs (CSphC #8) treated as indicated. Data are mean ± S.D. of three independent experiments using two different CSphCs (CSphC #8, #9). Samples were run on the same gel and images were cropped only for the purpose of this figure. Source data are provided as a Source Data file. **g** Kinetic growth of CR-CSphCs treated as indicated. **h** Number of invading CR-CSphCs at 48 h, pretreated with V-ASC CM and the indicated neutralizing antibodies for 48 h. **i** Flow cytometry analysis of CD44v6 positivity in CR-CSphCs treated as indicated, for 48 h. For **g–i** data are mean ± S.D. of six independent experiments performed with 2 different CR-CSphC lines (#8 and #9). For (b–d and g-i) statistical significance was calculated using the unpaired two-tailed t test. **j**, CD44v6 expression in CD44v6⁻ and GFP-transduced CD44v6 + cells after 3 days of exposure to V-ASC CM. One representative of 6 independent experiments is shown. **k** NGF, BDNF, NTF3, and NTF4 mRNA expression levels on CD44v6⁻ and CD44v6 + cells. Results show mean ± S.D. of three independent experiments performed with enriched cells from two different CR-CSphC lines (CSphC #8, #9). **l** Lollipop plot showing NGF, BDNF, NT-3, and NT-4 production by the indicated cells treated as indicated. **m**, Invasion assay of RFP transduced ASCs, using the indicated cells/media as chemoattractant agents. Scale bars, 100 μm. **n** Number of invading ASCs in presence of the indicated cells/media as chemoattractant agents. For **l–n** data are mean ± SD of three independent experiments using CR-CSphCs from different patients (CSphC #1, #8, #9, #21).

related to a CMS associated with a metastatic potential. Confirming the prominent role of released proteins in triggering metastasis pathways, exposure of CR-CSphCs to V-ASC CM turned the transcriptomic profile from an epithelial/CMS2 pattern into a phenotype that resembles the mesenchymal CMS4 (Fig. 3g). Likewise, a global RNA-Seq transcriptome analysis of CR-CSphCs treated with V-ASC CM or medium alone showed 10 DEGs, associated with a CMS4 signature (Fig. 3h). Accordingly, IL-6 and HGF promoted the downregulation of epithelial (*CDX2*, *E-cadherin*), and upregulation of mesenchymal (*CXCR4*, *SLUG*, *TWIST*, *ZEB1, ZEB2*) markers. Moreover, in CMS2 cells treated with V-ASC CM, targeting of both IL-6 and HGF restored the basal levels of EMT-related genes (Supplementary Fig. 3e). In addition, the treatment of CMS2 CR-CSphCs with IL-6 and/or HGF recapitulated the invasive scenario observed in CMS4 CR-CSphCs, which display a great ability to colonize the liver and lung. Conversely, IL-6 and HGF targeting abolished the metastatic activity induced by VAT, restoring the nonmetastatic phenotype of CMS2 CR-CSphCs (Fig. 3i). Thus, these data lay the groundwork for a critical consideration of released proteins in the regulation of the molecular machinery involved in metastasis. Taken together our results demonstrate that the paracrine activity of V-ASCs drives the transition of CRC sphere cells toward a mesenchymal-like phenotype, endowing them with a metastatic potential.

**VAT governs the EMT through regulation of ZEB2 expression.** We next explored the metastatic molecular events enhanced by V-ASCs. Based on a global RNA-Seq transcriptome analysis of CR-CSphCs exposed to V-ASC CM compared to those treated with medium alone, the gene set enrichment analysis (GSEA) performed with the molecular signatures database (MSigDB) showed negative enrichment of genes associated with metabolic pathways, and positive enrichment for genes related to EMT program (Fig. 4a). About 50% of EMT-related DEGs in CMS2 CR-CSphCs exposed to V-ASC CM are similar to those expressed in CMS4 CR-CSphCs. This includes upregulation of zinc finger E-box binding homeobox (ZEB) transcription factors *ZEB1* and *ZEB2* in line with the capacity of released proteins to reprogram CMS2 CR-CSphCs toward a pro-metastatic phenotype (Fig. 4b). In particular, 15 genes were upregulated while six downregulated in both the CMS4 and CMS2 CR-CSphCs treated with V-ASC CM (Fig. 4c). GSEA based on the differentially expressed genes from qPCR computed by MSigDB displayed an enrichment of terms associated with tumorigenesis, cell proliferation, EMT

signaling pathways, and cancer stemness (Supplementary Fig. 3f). ZEB1 and ZEB2, whose upregulation is induced both at mRNA and protein levels by V-ASCs cytokines (Fig. 4d, e), have been reported to modulate miR-200 family members expression and regulate EMT in CRC[48].

Analysis of a large spectrum of miRNAs expression reveals that miRNA-200 family members, and particularly miR-200a, are downregulated in CMS2 CR-CSphCs following treatment with V-ASC CM, showing similar miRNA levels exhibited by CMS4 CR-CSphCs (Fig. 4f and Supplementary Fig. 3g). In CR-CSphCs treated with V-ASCs CM, analysis of the network of most differentially expressed miRNAs and their targets showed that ASC-released proteins regulate a plethora of genes that converge on miR-200a/ZEB members axis involved in the EMT process (Fig. 4g). These data suggest that obesity-released proteins, through the modulation of miR-200a, could play a crucial role in determining the switch from an epithelial- to a mesenchymal-like phenotype, ultimately providing a considerable impact on disease progression. In line with this hypothesis, blockade of IL-6 and HGF restored the basal levels of miR-200a expression in CMS2 CR-CSphCs treated with V-ASC CM (Supplementary Fig. 3h). Moreover, CMS2 cells transfected with antagomiR-200a significantly increased their invasive capacity, whereas synthetic miR-200a led to a reduced number of invasive CMS4 cells despite their intrinsic mesenchymal-like phenotype (Supplementary Fig. 3i, j). To assess the effect of miR-200a in CR-CSphC behavior, we transduced CMS4 cells with miR-200a. CMS4 CR-CSphCs overexpressing miR-200a showed abrogation of *ZEB2* expression coupled with a significant reduction of both the basal as well as IL-6 and HGF-induced clonogenic and invasive potential (Supplementary Fig. 3k-o). Thus, we evaluated whether ZEB1 and ZEB2, under the control of V-ASCs secreted proteins, could contribute to the induction of EMT in CR-CSphCs. Both ZEB1 and ZEB2 are known to actively participate in the metastatic process and to be regulated by the miR-200 family through a negative feedback loop[49]. Importantly, the expression of both ZEB1 and ZEB2 have been associated with poor prognosis in CRC[50,51]. We therefore explored whether the expression of these EMT-inducing transcription factors influenced the metastatic capacity of CMS2 CR-CSphCs. ZEB2 protein expression levels in CMS2 cells exposed to V-ASC CM were similar to those detected in CMS4 cells (Fig. 4h). Moreover, exogenous expression of ZEB2 fostered expression of both vimentin and CD44v6 and enhanced the tumor spreading into distant organs (Fig. 4i, j, and Supplementary Fig. 4a–c), indicating that ZEB2 may act as a

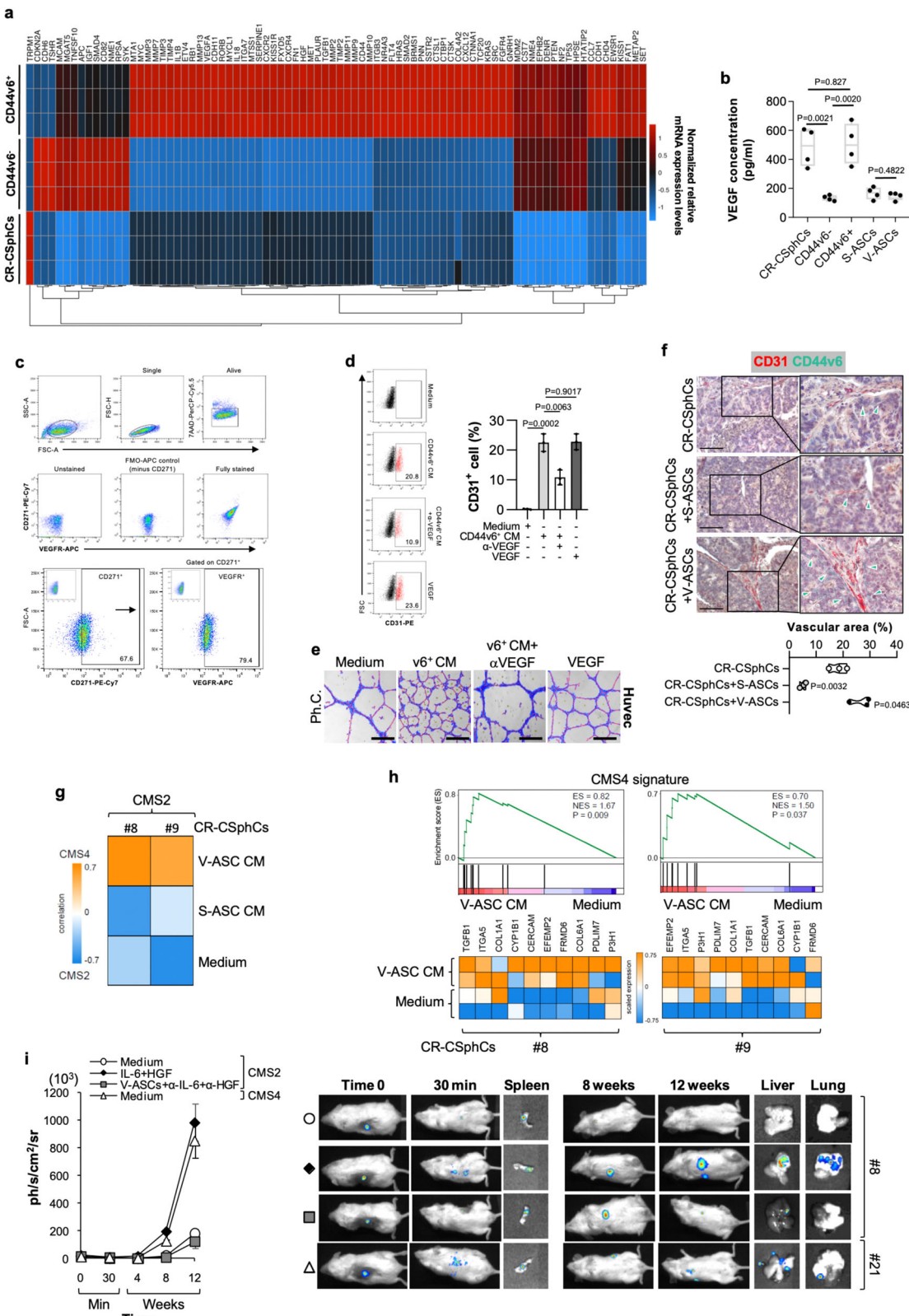

functional marker of CRC cells endowed with metastatic properties[52]. In accordance with these data, the analysis of a cohort of 37 tissue samples displayed a positive correlation between CD44v6 expression and CRC patients affected by obesity (Supplementary Fig. 4d). Knock-down of ZEB2 in CMS4 CR-CSphCs led to downregulation of vimentin and EMT-related genes and impaired the metastatic potential of these cells

(Supplementary Fig. 4e–i). Altogether, these results posit ZEB2 as a functional biomarker for CRC endowed with a metastatic potential.

**IL-6/HGF blockade reduces the metastasis formation induced by VAT.** To further define the signaling pathway involved in the regulation of miR-200a by pro-metastatic factors, we analyzed the

**Fig. 3 VEGF induces endothelial differentiation of ASCs, which activate the EMT program of CRC sphere cells. a** Clustergram of tumor microenvironment-related genes in CR-CSphCs (CSphC #1, #8, #9, #21) and CD44v6 − or CD44v6 + enriched cells. Data are presented as normalized expression values. **b** VEGF production in cells as indicated. Data are mean ± SD of 4 independent experiments. Box and whiskers show min-to-max values, with line indicating the mean value. **c**, Gating strategy of CD271/VEGFR expression on ASCs *(upper panels)*. Dot-plots of CD271/VEGFR staining with or without the indicated antibody (FMO-APC control, minus CD271-PE-Cy7) *(middle panel)*. Flow cytometry analysis of CD271 and VEGFR in ASCs. Data are representative of 3 independent experiments performed with 10 different ASC lines *(lower panel)*. **d**, Percentage of CD31 positivity, by flow cytometry analysis, on CD34 + /CD31⁻/CD45⁻ enriched ASCs exposed to vehicle (Medium), CD44v6 + CR-CSCs CM (CSC #1, #8, #9, #21), in presence or absence of VEGF neutralizing antibody, or VEGF for 14 days. Data are mean ± SD of three independent experiments using 3 different ASC cultures. **e** Phase-contrast micrographs of capillary-like tubular structures of Huvec cells treated as indicated for 16 h. Scale bars, 500 μm. One representative of 3 independent experiments is shown. **f** Immunohistochemical analysis of CD31 (red) and CD44v6 (green) on tumor xenografts generated by subcutaneous injection of CR-CSphCs alone or in combination with S-ASCs or V-ASCs *(upper panel)*. Percentage of vascular surface area, based on CD31 positivity, in tumor xenografts *(lower panel)*. Scale bars, 200 μm. Data are representative of 3 independent experiments. For **b, d,** and **f** statistical significance was calculated using the unpaired two-tailed *t* test. **g** Transcriptomic profile correlation between CMS2 CR-CSphCs (CSphC #8, #9) treated with S-ASCs or V-ASC conditioned medium (CM) and CMS4-associated gene signature. **h** GSEA of CMS4-associated gene signature in CMS2 CR-CSphCs (CSphC #8, #9) treated with V-ASC CM *(upper panel)*. Top ten significantly up- and downregulated CMS4 signature genes in treated cells *(lower panel)*. Statistical significance between two groups was determined by unpaired Student's t test (2-tailed). **i**, Kinetics and whole-body in vivo imaging analysis of mice ($n = 6$) intrasplenically injected with LUC-GFP CMS4, or CMS2 CR-CSphCs alone or co-injected with V-ASCs and treated as indicated. Data are mean ± S.D. of independent experiments performed with CR-CSphCs isolated from two different CMS2 (CSphC #8, #9) and CMS4 (#1, #21) CRC patients.

transcriptomic and the post-translational reprogramming of CR-CSphCs in presence of V-ASC CM. GSEA revealed that V-ASC-released factors promoted the enrichment of genes associated with STAT3-activated pathways in CR-CSphCs (Fig. 5a). Immunoblot analysis showed that both V-ASC CM and IL-6/HGF activate STAT3 pathway in CMS2 cells by enhancing the phosphorylation of STAT3 in tyrosine and serine residues (Fig. 5b). In accordance, immunohistochemical analysis of colon tumor sections indicated that activation of STAT3, highlighted by nuclear staining, is mainly located in proximity to adipose tissue of CRC patients with obesity. While cancer cells in contact with tumor stroma in lean patients displayed a weak presence of nuclear p-STAT3 (Fig. 5c). C188-9, an inhibitor of STAT3 phosphorylation, reduced the activation of STAT3, sustained by IL-6 together with HGF (Fig. 5d and Supplementary Fig. 5a). Moreover, in presence of IL-6 and HGF, C188-9 significantly lessened the cell proliferation rate, together with the clonogenic activity, and restored both miR-200a and ZEB2 expression levels in treated CR-CSphCs (Fig. 5e–h). Likewise, CMS2 CR-CSphCs, exposed to IL-6 and HGF, changed their cell morphology acquiring a distinctive polarization associated with an elongated shape, which is attenuated by the exposure to C188-9 (Fig. 5i). These results suggest that adipose microenvironmental cytokines, via STAT3/miR-200a/ZEB1/2 axis, drive epithelial-associated CMS2 CR-CSphCs to acquire a mesenchymal phenotype.

To circumvent this potentially detrimental activity of adipose tissue, we next analyzed whether therapeutic targeting of adipose-released proteins could prevent metastasis formation. Given that CR-CSphCs express IL-6R and c-MET, we focused on the currently available clinical compounds tocilizumab and crizotinib, which target the IL-6 and HGF pathway, respectively (Supplementary Fig. 6a). Importantly, the combinatorial treatment with tocilizumab and crizotinib was able to revert the impact of V-ASC CM on CR-CSphCs proliferation and colony-forming capacity (Fig. 6a, b). Furthermore, this dual-target therapy impeded the V-ASC-induced downregulation of miR-200a, thus reducing the expression of both ZEB1 and ZEB2, and the acquired invasive behavior of CMS2 CR-CSphCs (Fig. 6c–e). To determine whether tocilizumab, in combination with crizotinib, could be employed in an adjuvant setting and prevent metastasis formation induced by visceral adiposity, metastatic mouse avatars were generated by co-injection of CMS2 CR-CSphCs and V-ASCs into mice spleen. Five days after the splenectomy, mice were treated three times a week for 3 weeks (Fig. 6f). This optimized combinatorial drug dosage does not

provoke preclinical signs of toxicity as evidenced by the absence of significant body weight oscillations (Supplementary Fig. 6b). Strikingly, this therapeutic regimen significantly reduced the metastatic engraftment frequency of V-ASC-stimulated CMS2 cells, even 8 weeks after treatment suspension (Fig. 6g and Supplementary Fig. 6c, d). Moreover, in line with our in vitro data, we observed that most CRC cells found in metastatic lesions of mice co-injected with CR-CSphCs and V-ASCs showed mesenchymal traits highly expressing p-STAT3 and ZEB2 (Supplementary Fig. 6e). Furthermore, the analysis of a cohort of 112 CMS2 CRC patients showed a significant negative correlation between ZEB2 expression and RFS probability (Fig. 6h, i), suggesting that ZEB2 is a putative prognostic factor that could be relevant in cancer patients affected by obesity.

## Discussion

Here, we describe the paracrine role of VAT that through the activation of ZEB2 confers to CRC cells, endowed with an epithelial phenotype, a mesenchymal-like trait coupled with the ability to migrate and engraft at the distant site. CMS classification of CRC, based on gene expression analysis, has been recently proposed as a clinical tool to stratify patients according to the biological tumor behavior[35]. Among the distinct cancer subtypes, as a consequence of transition phenotype or tumor heterogeneity, a relevant fraction is represented by indeterminate or mixed nonconsensus samples[35]. Chronic adipose-derived proteins released in TME of patients with obesity, reprogram CMS2 CRC cells into a cell phenotype that is characterized by the partial expression of genes associated with CMS4 signature, likely reflecting a transient CMS2/CMS4 subtype. This phenomenon is in line with the presence of CRC patients classified as mixed, whose clinical outcome remains undefined and transcriptomic profile may reside in that category named "hybrid," "partial," or "reversible" epithelial-mesenchymal (E/M) phenotype[53,54]. Induction of E/M state ensures the reaction of cancer cells to microenvironmental stimuli preserving cancer stemness and tumor initiation abilities[55].

We predict that the future availability of integrated transcriptomic and clinical data may allow to define a new signature specific for CRC patients affected by obesity and to determine the prognostic impact of obesity in CMS2 CRC patients.

Our data reveal that V-ASCs are prominently present in a primary and metastatic lesion of CRC from patients with obesity. These cells release IL-6 and HGF, which in turn induce EMT up-regulating STAT3 phosphorylation and ZEB2 and increase the

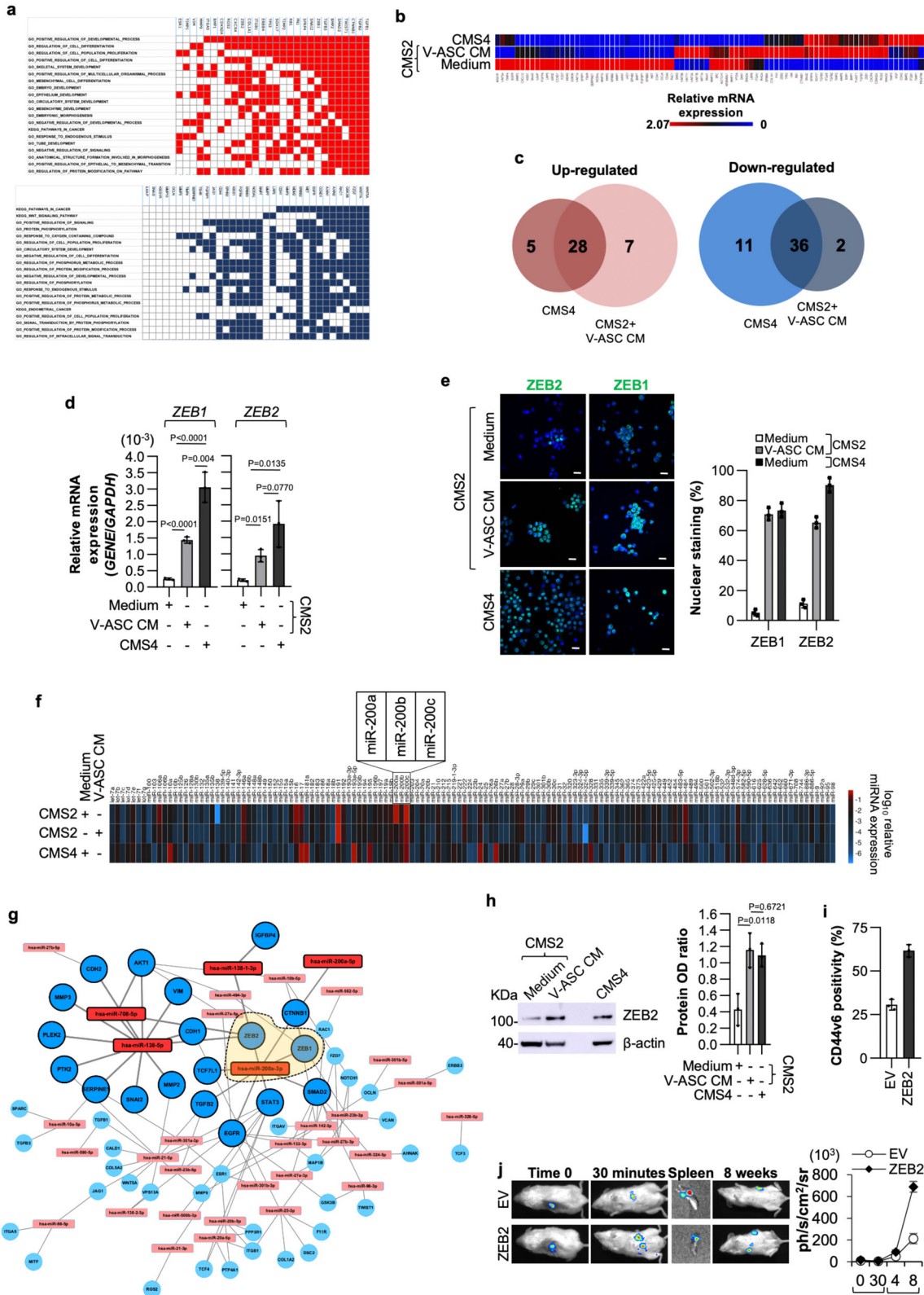

number of cells expressing CD44v6, the cell population able to colonize the liver and produce CRC metastases[43]. CD44v6 + cells produce NGF and NFT3, which in turn recruit ASCs promoting a paracrine loop that increase the tumor cell aggressiveness. The presence of adipose tissue in liver CRC metastasis is likely due to recruitment from disseminated CD44v6 cells and the migration capacity of V-ASCs in patients affected by obesity. In parallel with metastatic dissemination, VEGF released by CD44v6 + CRC cells likely contributes to increase tumor angiogenesis via the trans-differentiation of ASCs in endothelial cells and angiogenic sprouting of pre-existing vessels, thus sustaining the remodeling of tumor vasculature[56–58] (Fig. 6j). Although both IL-6 and HGF activate STAT3 and induce ZEB2, they seem to play a different role in tumor progression. HGF promotes cancer stemness and

**Fig. 4 V-ASCs enhance the expression of ZEB2 sustaining the metastatic activity of CR-CSphCs. a** Up- (red) and down- (blue) regulated genes and their relative top twenty significantly enriched gene sets (FDR q value ≤ 0.05), common in CMS4 CR-CSphCs (CSphC #1, #21) and CMS2 cells (CSphC #8, #9) treated with V-ASC CM, selected from all gene sets within MSigDB (H, CP Biocarta, CP Kegg, MIR, CGN, CM, BP, CC, MF, C6, C7). **b** Heatmap of EMT-related genes in CMS2 (CSphC #8, #9) and CMS4 (CSphC #1, #21) CR-CSphCs treated for 48 h as indicated. **c**, Venn diagrams of up- and downregulated genes in untreated CMS4 (CSphC #1, #21) and V-ASC CM-treated CMS2 (CSphC #8, #9) CR-CSphCs, compared to untreated CMS2 cells. **d** *ZEB1* and *ZEB2* mRNA expression levels in CMS4 and CMS2 CR-CSphCs treated as indicated. **e** Immunofluorescence analysis of CR-CSphCs expressing nuclear ZEB1 and ZEB2 (CMS2 #8, CMS4 #21) treated as indicated. Nuclei were counterstained with Toto-3. Scale bars, 20 μm. For **d** and **e** data represent mean ± S.D. of three independent experiments using CMS2 (#8, #9) and CMS4 (#1, #21) CR-CSphC lines. **f** Global gene expression profile of miRNAs in CMS4 (CSphC #1, #21) and CMS2 (#8, #9) CR-CSphCs treated as indicated. **g** Network of most differentially expressed miRNAs and their targets inferred from miRTarBase in CMS2 CR-CSphCs (CSphC #8, #9) treated with V-ASCs CM for 48 h. Bold colors represent miRNAs with a fold-change >8. Orange area within dashed line highlights *ZEB1* and *ZEB2* as direct targets of miR-200a. **h** Immunoblot analysis of ZEB2 in CMS4 (CSphC #1, #21) and CMS2 (CSphC #8, #9) CR-CSphCs treated as indicated. Data are mean ± S.D. of three independent experiments performed with CR-CSphCs isolated from 2 different CMS2 (CSphC #8, #9) and CMS4 (CSphC #1, #21) CRC patients. Samples were run on the same gel and images were cropped only for the purpose of this figure. Source data are provided as a Source Data file. For (**d** and **h**) statistical significance was calculated using the two-tailed *t* test. **i** Flow cytometry analysis of CD44v6 in CMS2 CR-CSphCs (CSphC #8, #9) transduced with empty vector (EV) or *ZEB2* synthetic gene. Bars represent means ± S.D. of three independent experiments using two CR-CSphCs. **j** In vivo whole-body imaging analysis of mice (n = 6) following intrasplenic injection of CR-CSphCs transduced as in **i** at 30 min and 8 weeks after splenectomy *(left panel)*. Luciferase signal measured as ph/s/cm$^2$/sr *(right panel)*. Data are mean ± S.D. of independent experiments performed with two CMS2 CR-CSphCs (#8, #9).

increases the clonogenicity, whereas IL-6 mainly contributes to enhance the invasion of CMS2 CR-CSphCs. In accordance with recent findings, the GSEA shows that the enrichment for genes associated with EMT and metastatic signature is dictated by the presence of VAT-released factors[59].

CMS2 CRC cells in contact with VAT-proteins activate STAT3 and up-regulate ZEB2 along with a decrease of miR-200a expression levels. Notably, over-expression of miR-200a hampers the invasive ability of CRC cells with a CMS4 signature, indicating that their constitutive metastatic behavior relies on signaling regulated by miR-200a/ZEB2 axis[60] (Fig. 6i). ZEB2 and miR-200a act through a mutual inhibitory feedback loop, which has been reported to be involved in the establishment of E/M cancer phenotype[55,61]. ZEB2 emerges as a prognostic predictive biomarker for CMS2 CRC, as confirmed by the inverse correlation between relapse-free survival and ZEB2 expression, which could be also relevant to identify cancers endowed with E/M traits and associated with high aggressiveness[55]. Thus, evaluation of ZEB2 expression levels could ameliorate CRC patient stratification for adjuvant therapy.

Despite the excess of AT accumulation directly influences tumor response to standard therapies due to augmented clearance and altered pharmacokinetics[62], the decreased overall survival of cancer patients with obesity could be also dependent on the molecular mechanisms governed by microenvironmental adipose-released proteins. Here we show that targeting the activation of the JAK/STAT pathway, by inhibiting STAT3 (C188-9), IL-6R (tocilizumab), or c-MET (crizotinib), impairs the VAT-driven metastasis promoting activity of CMS2 CR-CSphCs (Fig. 6i). Thus, because adipose-released factors fuel CRC microenvironment by determining a chronic inflammatory state and increasing the risk of distant metastasis formation[63], the clinical availability of these drugs could be taken into consideration for an adjuvant setting strategy in obesity-related cancer patients.

## Methods

**Tissue collection, isolation, and culture of cancer and adipose cells.** Colorectal cancer and adipose specimens were collected from CRC patients who underwent surgical resection, in accordance with the ethical policy of the University of Palermo Committee on Human Experimentation. Isolation and propagation of CR-CSphCs and ASCs were performed as previously reported[43,64]. CR-CSphC lines #1, #8, #9, and #21 were isolated from lean CRC patients. Adipose stromal cells from VAT and SAT were obtained from the greater omentum and subcutaneous anterior abdominal wall, respectively. The study received ethical approval for the purification and culture of CR-CSphC and ASCs, by Ethics

Committee 1 board, University of Palermo - Azienda Ospedaliera Universitaria "Paolo Giaccone" (authorization CE 6/2015). The study complied with all the ethical regulations for work with human participants, including obtaining the informed consent for both CRC patients with healthy weight, and affected by obesity.

Human samples were crosscut into small pieces and grinded using scalpels and surgical scissors and digested at 37 °C for 30 min in DMEM medium supplemented with 0.6 mg/ml of collagenase (Gibco) and 10 μg/ml of hyaluronidase (Sigma). The cell pellet was resuspended: i. for CRC, in serum-free stem cell medium (SCM) supplemented with EGF and b-FGF; ii. for adipose tissue, in mesenchymal stem cell medium (ThermoFisher Scientific), in ultra-low attachment cell culture flasks, leading to cell growth as spheroids. When cancer and adipose spheroids reached approximately 80% of confluence, cells were mechanically and enzymatically disaggregated, using Accutase (ThermoFisher Scientific). ASCs were routinely frozen and stored in liquid nitrogen at early passages to maintain their pluripotency (passage 1–12). Differentiated adipose cells were obtained by exposing ASCs, for up to 28 days, to adipogenesis differentiation medium (ThermoFisher Scientific). Differentiation efficiency was evaluated by AdipoRed assay (Lonza). Colorectal cancer sphere-derived adherent cells (SDACs) were obtained by culturing cells in adherent condition, in presence of 10% FBS, as previously described[41,65]. Huvec cells prescreened for angiogenesis were purchased by Lonza (C2519AS) and cultured according to the manufacturer's instructions.

CR-CSphC and ASC lines were routinely authenticated by short tandem repeat (STR) analysis using a multiplex PCR assay, including a set of 24 loci (GlobalFiler™ STR kit, Applied Biosystem), by comparing them to the parental patient tissues[66]. Conditioned medium was collected 48 h after cells reached subconfluence in SCM. The viability of CR-CSphC culture, which is routinely estimated by trypan blue and 7-AAD, is 93 ± 7%.

CR-CSphCs were treated with recombinant IL-6 (2 ng/ml; Novus), HGF (10 ng/ml; Peprotech), VEGF (10 ng/ml; Novus) and exposed to neutralizing antibodies against IL-6 (100 ng/ml; R&D), HGF (200 ng/ml; R&D), NGF (0.2 μg/ml; R&D), CD271 (0.5 μg/ml; Merck), and VEGF (10 ng/ml; R&D), tocilizumab (10 μg/ml; Selleckchem) or crizotinib (30 nM; Selleckchem), or to STAT3 inhibitor C188-9 (10 μM; Selleckchem). CM, cytokines, and neutralizing antibodies were added every 48 h to the cell culture.

Cell viability assay was performed using the CellTiter 96® AQueous One Solution Cell Proliferation Assay (MTS) according to the manufacturer's instructions and analyzed by using the GDV MPT reader (DV 990 BV6). To monitor the acquisition of epithelial versus a mesenchymal phenotype, $5 \times 10^3$ viable CR-CSphCs were embedded in 1:10 SCM/Matrigel solution and seeded as a single drop in a pre-warmed 24 well plate. Following matrigel polymerization, 500 μl of SCM were overlaid to each well and 3D organoid formation was followed up to 21 days.

**ELISA cytokines quantification.** Tumor inflammation, cell proliferation, and immune response cytokines, produced by cancer and adipose cells, were quantified by using the Bio-Plex Pro™ Human Cytokine 21-plex and 27-plex Assay (Bio-Rad), or by customized Luminex Assay specific for VEGF, NGF, BNDF, NT-3, NT-4 (R&D). Raw data were analyzed by Bio-Plex Manager Software 6.2 (Bio-Rad).

**Cell transfection and lentiviral transduction.** $1 \times 10^6$ viable CR-CSphCs were transfected with 8 μg of antagomiR-200a (ABM), or synthetic miR-200a (ABM), using the X-tremeGENE HP DNA Transfection Reagent (Roche) according to the manufacturer's instructions. Lentiviral particles were generated by transfecting

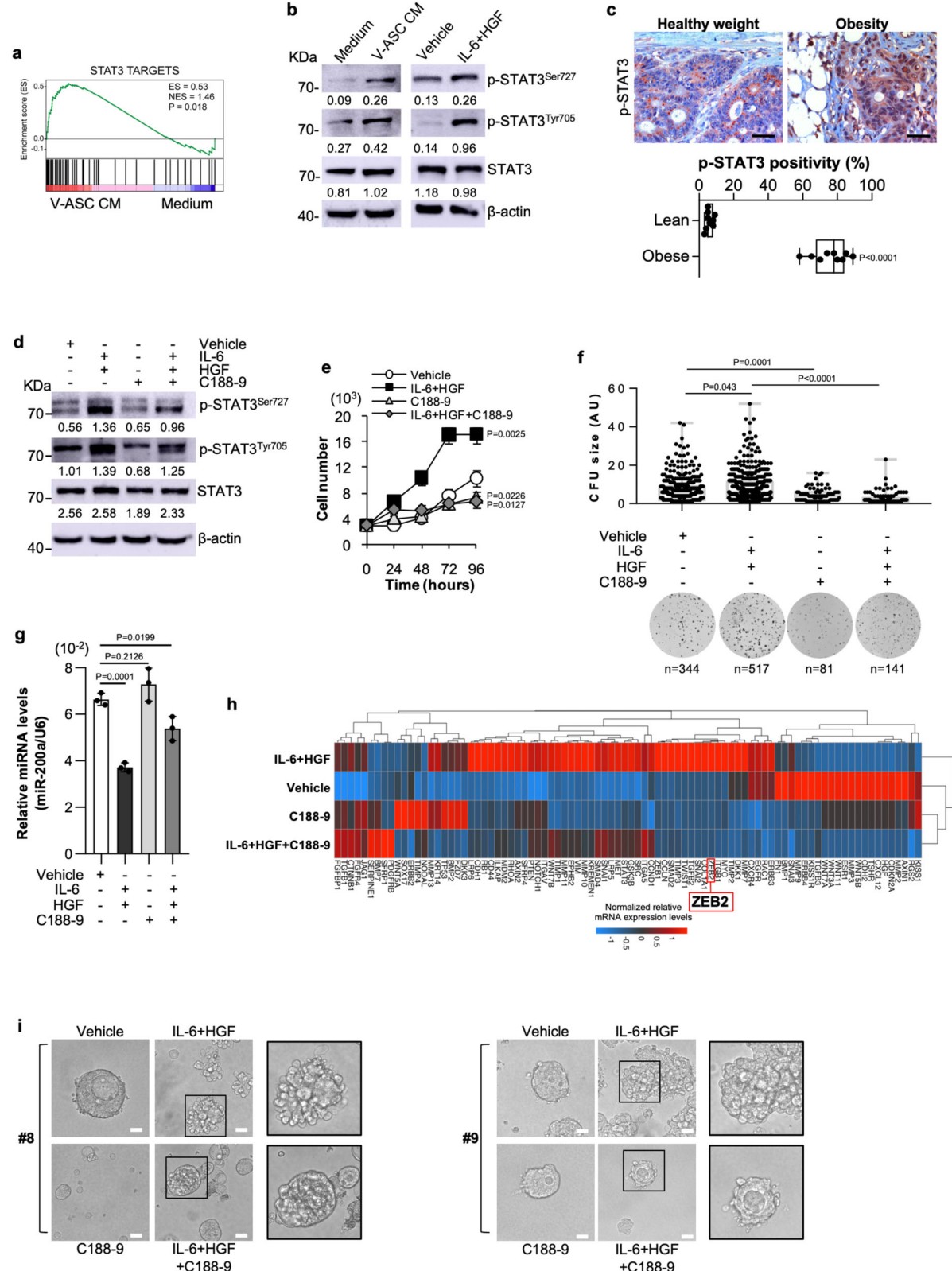

HEK-293 (ATCC, CRL-1573) packaging cells with TOP–GFP (Addgene), p-TWEEN LUC-GFP, pLenti-GIII-CMV-RFP-2A-Puro (ABM), ZEB2-RFP (ABM), nonsilencing shRNA control (Dharmacon), TRIPZ Human ZEB2 shRNA (Dharmacon), pLenti-III-mir-GFP-Blank (ABM) or LentimiRa-GFP-hsa-mir-200a (ABM) plasmids, together with psPAX2 (Addgene), and pMD2.G (Addgene) in OPTIMEM (Gibco) supplemented with XtremeGENE HP DNA Transfection Reagent (Roche). Cell transduction was fulfilled in presence of 8 µg/ml of polybrene (Sigma-Aldrich). Selection of resistant clones, where suitable, was performed by

treating cells with puromycin (1 µg/ml) for 5-10 days. Inducible gene expression was obtained by treating cells for 72 h with Doxycycline (1 µg/ml; Sigma).

**Immunohistochemistry/immunofluorescence and flow cytometry.** Immuno-histochemical analysis was performed on 4 µm-thick paraffin-embedded tumor sections, cytospins, or cells grown on glass coverslip, using a mix made by 100 µl of Antibody Diluent (Dako, S3022) and specific antibody against CDX2 (AMT28;

**Fig. 5 IL-6 and HGF induce a mesenchymal phenotype by activation of STAT3. a** GSEA of STAT3 targets gene signature in CMS2 CR-CSphCs (CSphC #8, #9) exposed to V-ASC CM. Statistical significance was calculated as described in Subramanian et al. (doi: 10.1073/pnas.0506580102). **b** Immunoblot analysis of phosphorylated STAT3 (p-STAT3$^{SER727/TYR705}$) and STAT3 in CMS2 CR-CSphCs (CSphC #9) treated as indicated. β-actin was used as a loading control. Samples were run on the same gel and images were cropped only for the purpose of this figure. Source data are provided as a Source Data file. One representative of three independent experiments is shown. **c** Immunohistochemical analysis of p-STAT3 on paraffin-embedded sections of CRC patients with healthy weight or affected by obesity. One representative experiment of nine is shown. Scale bars, 100 μm. Box and whiskers show min-to-max values, with a line indicating the mean value. **d** Immunoblot analysis of CMS2 CR-CSphCs (CSphC #9) treated as indicated. β-actin was used as a loading control. One representative of four independent experiments is shown. Samples were run on the same gel and images were cropped only for the purpose of this figure. Source data are provided as a Source Data file. **e** Proliferation rate of CMS2 CR-CSphCs exposed to the indicated treatment. Data are mean ± S.D. of three independent experiments using two different CR-CSphC lines (CSphC #8, #9). **f** Colony-forming analysis of CR-CSphCs treated as indicated, at 21 days. Boxes and whiskers represent mean ± S.D. of colony size performed in four independent experiments using four different CR-CSphC lines (CSphC #1, #8, #9, #21). *n* represents the number of colonies. **g** miR-200a expression levels in CMS2 cells treated as indicated using two different CR-CSphC lines (CSphC #8, #9). U6 was used as housekeeping control gene. Histograms represent mean ± S.D. of three independent experiments. For (c and e-g) statistical significance was calculated using the two-tailed *t* test. **h** Clustergram of stemness-related genes in CMS2 CR-CSphCs (CSphC #8, #9) treated for 48 h as indicated. *GAPDH* and *HPRT1* were used as housekeeping control genes. **i** Phase-contrast analysis of CMS2 CR-CSphCs grown in matrigel and treated as indicated at 10 days. One representative of four independent experiments carried out with two different CR-CSphC lines (CSphC #8, #9) is shown. Scale bars, 20 μm. For (d-i) STAT3 inhibitor C188-9 was used at 10 μM concentration.

mouse IgG$_{1κ}$, Novocastra, 1:50 dilution), Adiponectin (19F1; mouse IgG, ThermoFisher, 1:500 dilution), CK20 (Ks20.8, mouse IgG$_{2a,κ}$, Novocastra, 1:50 dilution), p-STAT3 (Tyr705; 9145, rabbit IgG, CST, 1:50 dilution), STAT3 (9139, mouse IgG$_{2a}$, CST, 1:300 dilution), and ZEB2 (E-11, mouse IgG$_{2a}$, Santa Cruz, 1:100 dilution). Single staining was revealed using biotine-streptavidine system (Dako) and detected with 3-amino-9-ethylcarbanzole (Dako). Double staining was performed using antibodies against CD31 (AB28364, rabbit IgG, Abcam, 1:50 dilution) and CD44v6 (2F10, mouse IgG1, R&D systems, 1:100 dilution), or using antibodies against CK20 (Ks20.8, rabbit IgG, CST, 1:100 dilution) and ki67 (D3B5, rabbit IgG, CST, 1:100 dilution), revealed by the MACH 2 double stain 2 kit conjugated goat antimouse polymer horseradish peroxidase (HRP) and the conjugated goat antirabbit polymer alkaline phosphatase (AP) (Biocare Medical), and detected by DAB or Vina Green, and Vulcan Fast Red chromogen. Triple staining was performed using antibodies against CD34 (ICO115, mouse IgG$_1$, CST, 1:50 dilution), CD31 (JC70A, mouse IgG$_{1κ}$, Dako, 1:50 dilution), and CD45 (D9M8I, rabbit IgG, CST, 1:200 dilution), revealed by specific secondary antibodies, and detected by DAB, Vina Green and Vulcan Fast Red chromogen, respectively. For paraffin-embedded sections of primary CRC, the staining was detected using the Fast Red/Diaminobenzidine (DAB) substrates, while Fast Red/Vina Green were used in liver metastasis to avoid the intrinsic brown color background. Nuclei were counterstained with aqueous hematoxylin (Sigma). H&E stainings were performed using standard protocols. The adipose tissue and the vascular area were evaluated by calculating the area covered by Adiponectin$^+$ and CD31 + cells, respectively (ImageJ, Colour deconvolution plugin). "Cell counter" plugin (ImageJ 1.8.0_172) was used to determine the number of cells expressing CD34/CD31/CD45, CK20/ki67, or p-STAT3. The analysis was assessed on five tumor sections of primary and metastatic tissue, for each experimental condition.

Retrospective immunohistochemistry score of CD44v6 expression in CRC patients with healthy weight and affected by obesity has been measured as quick score (*Q*), by multiplying the percentage of positive cells (*P*) and the intensity (*I*).

For immunofluorescence, cells were cytospun or grown on glass coverslips, fixed, permeabilized, and exposed overnight at 4 °C to a mix made by 100 μl of 3% BSA 0,05% Tween-20 PBS and specific antibodies against ZEB2, ZEB1 (H-102, rabbit IgG, Santa Cruz, 1:100 dilution), WT1 (83535, rabbit IgG, CST, 1:100 dilution), IL-6R (PA5-47209, goat IgG, ThermoFisher Scientific, 1:50 dilution), c-Met (95106, mouse IgG$_1$, R&D, 1:100 dilution), or isotype-matched control (IMC), as previously described[43]. Primary antibody staining was detected by using antirabbit, antigoat, or antimouse secondary antibodies conjugated with Alexa Fluor-488 or Rhodamine Red-x (Life Technologies). Nuclei were counterstained using Toto-3 iodide (Life Technologies), or DAPI (ThermoFisher Scientific). Lipid droplets were stained using the Oil Red O (Sigma).

For flow cytometry analysis, following antibody titration, $1 \times 10^5$ cells were collected, washed in PBS and stained for 1 h at 4 °C with conjugated antibodies specific for CD44v6 (2F10 APC, mouse IgG$_1$, R&D systems, 5 μl/sample), CD271 (C40-1457 PE-Cy7, mouse IgG1κ, BD, 12 μl/sample), VEGFR (89106 APC, mouse IgG$_1$, R&D, 10 μl/sample), CD31 (WM59 PE, mouse IgG$_{1κ}$, BD, 5 μl/sample), CD34 (581 APC, mouse IgG$_{1κ}$, BD, 20 μl/sample), CD45 (5H9, mouse IgG$_{1κ}$, BD, 5 μl/sample), CD90 (5E10 PE, mouse IgG$_{1κ}$, BD, 10 μl/sample), CD105 (166707 APC, mouse IgG$_1$, R&D, 10 μl/sample), CD73 (AD2, mouse IgG$_{1κ}$, BD, 5 μl/sample), CD10 (97C5 APC, mouse IgG$_{1κ}$, Miltenyi Biotech, 5 μl/sample), CD200 (MRC OX-104 APC, mouse IgG$_{1κ}$, BD, 20 μl/sample), CD36 (CLB-IVC7, mouse IgG$_{1κ}$, BD, 5 μl/sample), CD106 (51-10C9, mouse IgG$_{1κ}$, BD, 5 μl/sample), or corresponding IMC. Dead cells' exclusion was performed by using 7-AAD (0.25 μg/$1\times10^6$ cells, BD Biosciences).

Isolation of TOP–GFP$^{high/low}$ and CD44v6 +$^{/-}$ CR-CSphCs, or CD34 + / CD31$^-$/CD45$^-$ ASCs was performed by using the FACSMelody cell sorter. Cells were washed with 2% BSA and 2 mM EDTA PBS and filtered with a 70 μM mesh to prevent cell clogging. Postsorting analysis was performed to verify the purity of the sorted population. Dead cells' exclusion was performed by staining cells with 7-AAD.

**Clonogenesis, colony forming, and invasion assay**. Clonogenicity of bulk or enriched Wnt$^{high/low}$ CR-CSphCs, was determined by plating 1, 2, 4, 8, 16, 32, 64, 128 cells per well, in medium or V-ASC CM, and analyzed with the Extreme Limiting Dilution Analysis (ELDA) 'limdil' function (http://bioinf.wehi.edu.au/software/elda/index.html).

CR-CSphCs were seeded as dissociated cells ($2 \times 10^4$ viable cells) in 0.3% agar (SeaPlaque Agarose; Lonza) and cultured up to 21 days. Colony-forming potential was assessed by staining cells with 0.01% crystal violet in 1% methanol. Colonies were counted using ImageJ software based on the size (small <7 pixels, medium 7–13 pixels, and large >13 pixels).

The invasive potential was evaluated by seeding $2 \times 10^3$ viable CR-CSphCs in 200 μl of SCM into 8 μm pore size transwell coated with 30 μl of 1:6 Matrigel/SCM solution (BD Biosciences). SCM supplemented with 10% human serum was used as chemoattractant, alone or in combination with NGF (10 ng/ml; Peprotech), NGF neutralizing antibody (0.2 μg/ml; R&D), or CD271 neutralizing antibody (ME20.4, mouse IgG1, Merck). To study the paracrine role of CD44v6 + cells in the induction of an invasive phenotype on ASCs, $1 \times 10^5$ enriched viable cells were seeded in the lower chamber of 24 well plate until confluence was reached.

**Animal models and Treatments**. Six weeks old male NOD-SCID mice were purchased by Charles River Laboratories and housed at the University of Palermo in accordance to institutional guidelines of the Italian animal welfare (D.L. n° 26 March 4, 2014) and authorization n. 951/2015-PR. Mice were maintained in a temperature-controlled system (22 °C, 50% humidity) with a 12 h dark/light cycle, with *ad libitum* access to pelleted chow (Special Diets Services-811900 VRF1 (P)) and to 0.45 μm filtered water in sterile drinking bottles, in cages (Tecniplast) with radiation-sterilized bedding (SAWI Research Bedding, JELU-WERK).

CIC capacity was evaluated by injecting 50 μl of 1:3 Matrigel/SCM containing luciferase LUC/GFP-transduced CR-CSphCs (10, 100, 1000, or 10,000), alone or in combination with V-ASCs (50,000), in the subrenal capsule of NSG mice. Following i.p. injection of VivoGlo Luciferin (150 mg/kg), tumor growth and metastasis formation were monitored by evaluation of bioluminescence emission, which was detected by the whole-body imaging system (IVIS Lumina III, PerkinElmer). CIC frequency was analyzed with the Extreme Limiting Dilution Analysis (ELDA) 'limdil' function (http://bioinf.wehi.edu.au/software/elda/index.html).

Subcutaneous xenografts were generated by injecting 150 μl of 1:1 Matrigel/SCM solution containing $2.5 \times 10^5$ viable CR-CSphCs, alone or in combination with S-ASCs or V-ASCs ($1.25 \times 10^5$ viable cells). After tumor appearance (0.03–0.06 cm$^3$), mice were monitored twice a week and tumors were measured by using a digital caliper. Tumor volume was calculated using the formula largest diameter x (smallest diameter)$^2$ x π/6. Once the endpoints were reached, with subcutaneous tumors having the largest diameter = 2 cm, or when mice showed signs of suffering, animals were sacrificed accordingly to Directive 2010/63/EU guidelines (D.lgs 26/2014).

In vivo metastatic potential of CR-CSphCs was assessed by intrasplenic injection of 30 μl of PBS solution in presence of $3 \times 10^5$ luciferase LUC/GFP-transduced cells, alone or in combination with $1.25 \times 10^5$ V-ASCs. To ascertain the in vivo migratory/engraftment potential, $1.25 \times 10^5$ LUC-GFP-transduced ASCs were injected into NOD-SCID mice spleen. The spleen was removed 30 min after cell injection. Where indicated, mice were treated i.p. with tocilizumab (10 mg/kg) in combination with crizotinib (5 mg/kg) 3 days/week for 3 weeks. At 12 weeks,

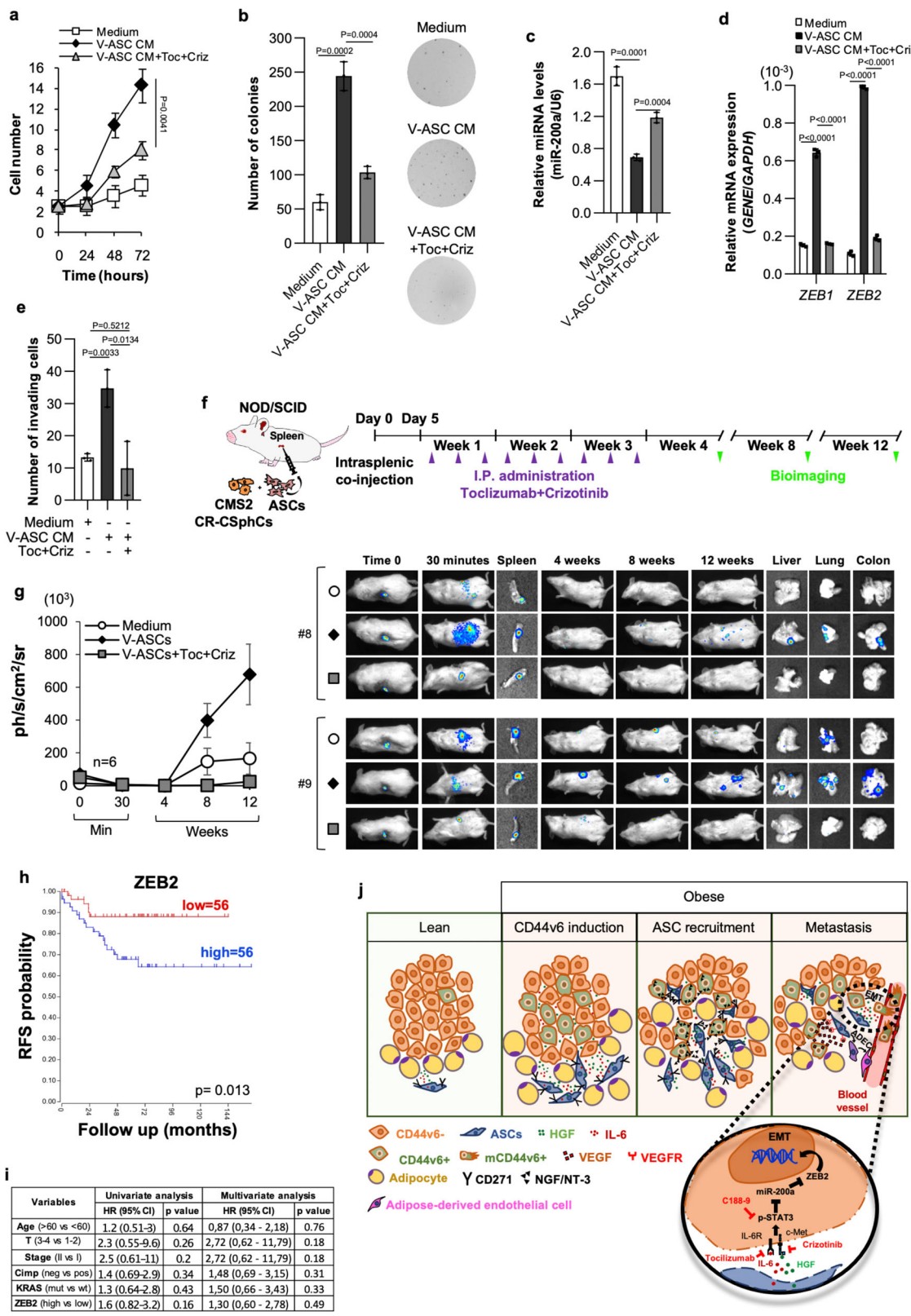

after mouse sacrifice, organs were harvested, macroscopically analyzed, and fixed in formalin for histological analysis.

**Tube formation assay**. Huvec cells were plated on matrigel coated wells (200 µl/cm²) at 80,000 cells/cm² seeding density with 170 µl/cm² of EGM-2 growth medium, in presence or absence of 10 ng/ml VEGF, or CD44v6 + CR-CSphCs' CM. Plates were incubated for 16 h at 37 °C in humidified incubator 5% $CO_2$, before microscopy observation. Phase-contrast images of Huvec-derived tubes were

captured using the EVOS microscope (AMG) at x4 magnification. Image analysis for the evaluation of tubule characteristics (tubules covered area (%), tubules length, total number of tubules, mean of tubules length) was performed using the Wimasis software (https://www.wimasis.com/en/).

**Real-time PCR and RNAseq**. RNA was retrotranscribed using the High-Capacity cDNA Archive Kit (Applied Biosystems). Quantitative Real-time PCR (qRT-PCR) was accomplished in a SYBR Green PCR master mix (Qiagen) containing primers

**Fig. 6 IL-6 and HGF targeting hampers the VAT-induced metastatic capacity of CR-CSphCs. a** Growth kinetics of CMS2 CR-CSphCs treated with V-ASC CM, alone or in combination with tocilizumab (Toc) and crizotinib (Criz). **b** Colony-forming assay of CR-CSphCs following the indicated treatment, at 21 days. **c**, miR-200a expression in CMS2 CR-CSphCs treated as indicated for 48 h. U6 was used as housekeeping control gene. **d** *ZEB1* and *ZEB2* expression levels in cells treated for 72 h as indicated. *GAPDH* was used as housekeeping control gene. **e** Number of invading CMS2 CR-CSphCs pretreated as indicated for 48 h. Statistical significance between two groups was determined by unpaired Student's t test (2-tailed). For (a-e) data are mean ± S.D. of three independent experiments using two different CR-CSphC lines (CSphC #8, #9). Statistical significance was calculated using the two-tailed *t* test. **f** Schematic model of intrasplenic injection of CR-CSphCs showing time points of treatments and in vivo bioluminescence detection. **g** Kinetics and whole-body imaging analysis of mice (*n* = 6) following intrasplenic injection of LUC-GFP-transduced CMS2 CR-CSphCs alone or co-injected with V-ASCs untreated or treated with the indicated pharmaceutical compounds. Insets represent spleen collected 30 min after cell injection, and liver, lung, and bowel collected at the time of sacrifice. Data are mean ± S.D. of independent experiments using two different CR-CSphC lines (CSphC #8, #9), and 2 S- (#3, #6) or V-ASC (#5, #14) lines. **h** RFS rate of CMS2/MSS/Stage1-2 CRC patients according to ZEB2 expression levels. Statistical significance was calculated using the log-rank (Mantel–Cox) test. **i** Univariate and Multivariate analysis of relapse-free survival (RFS) according to regression Cox model in CRC patients as in **h**. Statistical significance was calculated using the Wald test. **j** Schematic representation of bidirectional crosstalk between ASCs and CRC cells. Visceral adipose factors enhance the expression of CD44v6; CD44v6 + -released NGF/NT-3 drives the intra-tumor recruitment of ASCs; adipose-released proteins induce EMT of CRC cells through the activation of STAT3; CD44v6 + -released VEGF promotes the endothelial transdifferentiation of ASCs. Clinically available drugs targeting HGF and IL-6 are highlighted in red.

for *NGF, BDNF, NTF3, NTF4, ZEB1, ZEB2, CDX2, E-CADHERIN, CXCR4, FRMD6, N-CADHERIN, SLUG, SNAIL, TWIST,* and *VIMENTIN*. GAPDH was used as endogenous control. Primers sequences are included in Supplementary Table 3.

mRNA expression levels of EMT- (PAHS-090Z, Qiagen) and tumor metastasis-related genes (PAHS-028Z, Qiagen) were detected by $RT_2$ profiler PCR array. The gene expression profile of EMT-, Wnt- and CSCs-related gene expression was evaluated by using the PrimePCR Custom Panel (Bio-Rad) according to manufacturer's instructions. To evaluate miRNA expression levels, total RNA was retrotranscribed by using miScript reverse Transcription Kit (Qiagen). miRNA expression profile was determined using Megaplex pools kit (Applied Biosystem) specific for a set of 384 microRNAs (TaqMan Human MicroRNA Array A) as recommended by manufacturer's instructions. Collected data were analyzed with the Thermo FisherCloud software. miR-200a, miR-200b, and miR-200c expression levels were evaluated by qRT-PCR using specific primers (Qiagen). miRNA expression levels were normalized with endogenous RNU-6 (Qiagen) control and calculated using the comparative Ct method ($2^{-\Delta\Delta Ct}$). For measuring mRNA expression, the NEBNext Ultra Directional RNA Library Prep Kit for Illumina was used to process the samples. The sample preparation was performed according to the protocol 'NEBNext Ultra Directional RNA Library Prep Kit for Illumina' (NEB). Briefly, mRNA was isolated from total RNA using the oligo-dT magnetic beads. After the fragmentation of the mRNA, a cDNA synthesis was performed. This was used for ligation with the sequencing adapters and PCR amplification of the resulting product. The quality and yield after sample preparation were measured with the Fragment Analyzer. The size of the resulting products was consistent with the expected size distribution (a broad peak between 300 and 500 bp). Clustering and DNA sequencing using the Illumina NextSeq 500 was performed according to the manufacturer's protocols. A concentration of 1.6 pM of DNA was used. NextSeq control software 2.0.2 was used. Image analysis, base calling, and quality check were performed with the Illumina data analysis pipeline RTA v2.4.11 and Bcl2fastq v2.17. Raw sequencing reads were aligned to Ensembl release 84 (GRCh38 assembly) using the HISAT2 2.1.0 pipeline[67] and counts were summarized per gene using the summarizeOverlaps function in the GenomicsAlignment R package[68]. Gene counts were normalized to reads per million and log2 transformed. The CMS signature was derived from the TCGA dataset[69], which expression data were downloaded from the FIREHOSE repository (https://gdac.broadinstitute.org/). This included RNAseq expression data generated by the Illumina HiSeq (*N* = 326) and Genome Analyzer (*N* = 172) platforms (RSEM normalized data). After log2 transformation, data from both platforms were combined into a single dataset (*N* = 498), correcting platform-specific effects with the ComBat algorithm[70] as implemented in the sva R package[71]. As a first step toward defining a CMS signature, genes were selected with at least one read per gene in all 12 cell line expression profiles (*N* = 10,949). This set of genes was further filtered based on a correlation of correlations approach[35] comparing the TCGA dataset with the cell line expression data and retaining genes with a correlations of correlation score ≥0.1 (*N* = 4481). CMS labels for the TCGA samples were obtained from Guinney et al.[35] (134 CMS2, 81 CMS4) and differential gene expression (CMS4 versus CMS2) was determined using the limma *R* package[72]. The CMS signature was constructed using the log2 fold changes of the 10 most significantly upregulated and 10 most significantly downregulated genes. This signature was used to calculate the Pearson correlation coefficient with the cell line expression profiles, averaging the treatment replicates and mean-centering within the cell line (both gene-wise). GSEA[73] using the CMS4 signature, as well as the tumor stemness and STAT3 targets gene sets, was performed using the GSEA software (http://software.broadinstitute.org/gsea/index.jsp).

Gene set enrichment analysis was accomplished in R with fgsea package by considering all gene sets from MSigDB collections (msigdb.v6.2.symbols) with sample permutation (1,000 permutations). A differential expression analysis was performed with limma package.

**Western blot**. Cells were lysate in ice-cold buffer, loaded in SDS-PAGE gels, and blotted on nitrocellulose membranes. Membrane were pre-incubated with blocking buffer (0.1% Tween 20 and 5% nonfat dry milk in PBS) for 1 h at room temperature and then exposed to a mix made by 5% BSA 0,05% Tween-20 PBS and specific antibodies (1:1000 dilution) against ZEB2 (E-11, mouse $IgG_{2a}$, Santa Cruz), p-STAT3 (Ser727) (rabbit, CST), p-STAT3 (Tyr705) (9145, rabbit IgG, CST), STAT3 (9139, mouse $IgG_{2a}$, CST), active β-catenin (Ser33/37/Thr41) (D13A1, rabbit IgG, CST), β-catenin (D10A8, rabbit IgG, CST), vimentin (R28, rabbit, CST), or β-actin (8H10D10, mouse, CST). Primary antibodies were revealed using anti-mouse or antirabbit HRP-conjugated (goat H + L, ThermoFisher Scientific, 1:2000 dilution in blocking buffer) and detected by Amersham imager 600 (GE Healthcare). Protein levels were calculated by densitometric analysis using ImageJ software v1.8.0_172.

**Statistical analysis**. The meta-analysis was conducted after the selection of all the relevant studies on PubMed database related to BMI and survival of CRC patients. The studies were classified by year and population size, and the relative Hazard Ratio (HR), Risk Ratio, and Odds Ratio values were collected. In order to account for the missing confidence intervals (CI) for the estimate of effect in a study we used the method as proposed in Altman et al.[74]. Meta-analysis was conducted by using metafor package in R and choosing the default Restricted Maximum-likelihood Estimator (REML) estimation to fit the model.

Survival results were obtained from an internal database of medical records or using the "R2: Genomics Analysis and Visualization Platform (http://r2.amc.nl http://r2platform.com)" and analyzed with a log-rank (Mantel–Cox) test and expressed as Kaplan–Meier survival curves.

Clinical data for progression-free survival, univariate and multivariate analysis, were obtained from Istituto Oncologico del Mediterraneo (*n* = 393), and Azienda Ospedaliera Universitaria "Policlinico Vittorio Emanuele" (*n* = 441). Univariate and multivariate analysis were performed in R (v R-4.1.0) with "survival" package by fitting a Cox proportional hazards regression model to evaluate the risk of Progression-Free Survival (PFS) associated with each covariate. PFS was calculated with right-censored data and the event of distant recurrence for each patient.

Data were shown as mean ± standard deviation. Following Kolmogorov–Smirnov test to assess the samples distribution, statistical significance was estimated by unpaired *T* test, or by two-tailed Mann–Whitney test. Results were referred to statistically significant as $p < 0.05$. * indicates $p < 0.05$, ** indicate $p < 0.01$, *** indicate $p < 0.001$, and **** indicate $p < 0.0001$.

**Reporting summary**. Further information on experimental design is available in the Nature Research Reporting Summary linked to this paper.

## Data availability

All data relevant to the study are included in the article or uploaded as supplementary information. Uncropped western blots are included in the Source Data file. The data that support the findings of this study are available from the corresponding author (GS) upon reasonable request. RNA sequencing data of CR-CSphCs treated with control medium, S-ASC or V-ASC CM generated in this study have been deposited in a public, open-access GEO repository, under accession number GSE162561 (link to data:). The CMS signature was derived from the TCGA dataset, which expression data were downloaded from the FIREHOSE repository (https://gdac.broadinstitute.org/). CMS labels for the TCGA samples were obtained from Guinney et al. and are available at the website:

https://www.synapse.org/#!Synapse:syn2623706/files/, following login in. Source data are provided with this paper.

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

## Acknowledgements
We are thankful to Francesco Calo' for graphics. The research leading to these results has received funding from AIRC under 5 × 1000 (9979) project to Giorgio Stassi and Ruggero De Maria; ONCODE, Transcan-2 grant Tactic, Dutch Cancer Society (KWF) Grants UvA2015-7587 and 10150 to Jan Paul Medema; AIRC IG (21445) and PRIN 2017WNKSLR to Giorgio Stassi, RF2018-12367044 to Matilde Todaro and Ruggero De Maria.

Alice Turdo and Veronica Veschi are research fellows funded by European Union-FESR FSE, PON Ricerca e Innovazione 2014–2020 (AIM line 1).

## Author contributions
S.D.F. and G.S. conceived and designed the experiments; S.D.F., P.B., A.T., M.G., V.V., A.N., L.R.M., M.L.I., I.P., M.E.F., L.C., and M.T. carried out the experiments, analyzed and elaborated data; S.D.F., D.S.S., S.V.H., and E.M. executed the bioinformatics analysis; M.R.B., R.D.M., and J.P.M. supplied scientific suggestions and critical review; F.M., G.M., L.L., G.G., D.G., L.M., M.M., and P.V. provided clinical data of CC patients; M.T., R.D.M., J.P.M., and G.S. provided critical comments to the manuscript; S.D.F and G.S. wrote the manuscript; R.D.M., J.P.M., and G.S. provided funds.

## Competing interests
The authors declare no competing interests.
