## [Peer Review File · Nature Communications]

Adipose stem cell niche reprograms the colorectal cancer stem cell metastatic machineryREVIEWER COMMENTS

Reviewer #1 (Remarks to the Author); expert on cancer stem cells and colorectal cancer:

After reviewing the manuscript by Di Franco S et al entitled "Adipose stem cell niche reprograms the colorectal cancer stem cell metastatic machinery" we consider it a very interesting study. It provides some new insights on CRC biology and addresses several questions that could potentially be relevant for the management of patients. However, there are several aspects that should be improved prior its publication in Nature Communications.

MAJOR CONCERNS

1. The study lacks general novelty since the contribution of obesity and adipose tissue to cancer and CRC in particular has been well documented already. Despite this fact, the present study shows some new valuable data on the cellular/molecular mechanisms ruling such influence of obesity on metastatic cancer.
2. As a general idea, authors recurrently claim that cancer stem cells (CSC) are particularly relevant for the metastatic potential of CRC. It is well described that the gold standard definition of CSC is based on their tumor initiating capacity. However, authors do not show any tumor initiation assay in vivo to support their conclusions. This is fundamental to claim the role of CSC in this study. This type of assay require the injection of tumor +/- stromal cells in a limiting dilution manner in immunodeficient mice. We recommend injection of cells in the kidney capsule to help on tumor initiation and scape from the contribution of local adipose tissue. These experiments should be done to complement several parts of the study (Figure 1F, 1G; Figure 2C; Supplementary 2; Figure 5E, 5F; Figure 6A, 6B). Only in this manner, authors will prove the effect of V-ASC, or IL6+HGF +/- Inhibitory antibodies or small drug inhibitors (Toc+Criz) on CSC biology. We know that these are very demanding experiments and therefore an alternative solution would be to eliminate the stemness concept from the whole manuscript and just focus on those capacities of tumor cells really evaluated in the study such as proliferation, invasion and metastasis.
3. For most of the analyses performed on patient cohorts there is a serious lack of information. Authors must detail received treatments (adjuvancy?), curative surgery, and also perform a multivariate analyses including most relevant parameters that impact survival such as mutational status of KRAS, BRAF, PI3K; TNM, Grade, etc, in all studies with patient samples. This level of analysis is fundamental for concluding on the potential impact of the discoveries on CRC patients. Therefore, this should be provided for Figure 1B, Supplementary Figure 2F and Figure 6H.
4. Mutational profile is key for CRC progression and response to treatments. Authors should evaluate the impact of relevant oncogenic mutations for CRC on genes such as KRAS, BRAF, PIK3CA, SMAD4 and APC, on the response to the adipocytic niche. The use of small drug inhibitors of the KRAS/BRAF/MEK pathway will inform on their contribution on the Adipocyte/cancer cell crosstalk.
5. Authors should indicate the origin of tumoroids used in all the experiments (e.g. Figure 1F-I. Are they derived from a CRC carcinoma of lean or obese patients? Are they generated from a metastasis or a primary tumor? This should be indicated for all experiments across the manuscript.
6. Figure 1G. Quantification of proliferation in subcutaneous xenografts should be presented (e.g. Ki67 stainings) to strength the point of V-ASCs promoting tumor growth. Besides, a quantification of adipocytes should also be presented in the same xenografts. Are the injected adipocytes proliferating? Could adipocytic stroma impact on final tumor volume?
7. Figure 3F. The effect on vasculature should be quantified more precisely. In the case of tumor xenografts, a CD31 staining is recommended. Markers of ASCs should also be evaluated and quantified. It would also be important to show if the location of CD44V6+ cells is preferentially close to vascularized tumor areas (niches).
8. Figure 3G-I. Does IL-6+HGF treatment promote the transition from CMS2 to CMS4 signature? This gene expression profile transition is prevented by antibodies blocking IL-6 and or HGF in tumor cells exposed to V-ASC CM? This is relevant if authors want to conclude that this change in whole gene expression is ruled by IL-6 and/or HGF and required for the acquisition of metastatic potential.
9. Figure 3j and Suppl. Fig.4g: Please show quantification of luciferase signal in the in vivo experiment with ZEB1 overexpressing versus control cells.
10. Authors conclude that obese CRC patients would be enriched in tumors with CMS4. Is this the

case? Or are they CMS2 with some traits of EMT such as ZEB2 high expression? Do these potential subgroups present differential RFS or PFS?

MINOR

1. Figure 1C. A magnification of CDX2 staining in the metastatic tumors will be appreciated. The current image is not clear enough.
2. Figure 1D. The Leptin staining looks with high unspecific background. It would be much appreciated to present the results with another antibody marking adipocytes or an alternative technique (e.g. RT-PCR).
3. Figure 2. The conclusions on the CD44V6+ population would be very much strengthened by showing their increased presence in CRC carcinomas of obese patients.
4. Supplementary Figure 2a. The staining of nuclear beta-catenin is not convincing. Are these cells APC mutant? This should be quantified by western blot or other means. Nuclear beta-catenin accumulation is not a definitive marker of "stemness". PDOs from #8 should be also characterized for beta-catenin.
5. Figure 3d. Images are not informative and could be removed or substituted by a flow cytometry analysis of CD31. There is a label mistake on the quantification plot showing the cells treated with anti-VEGF antibody.
6. Figure 4a: Gene sets positively enriched by V-ASC-CM do not show clearly a relationship with EMT. We recommend to move Supplementary Figure 3d in to the main Figure 4 to better support this conclusion.
7. Figure 3b: Gene names are shown in a letter size too small. They cannot be read. Please magnify.
8. Figure 4e: Please quantify changes in ZEB1 and ZEB2 protein expression.

In summary, although this is an interesting study, there are many aspects to be significantly improved before its publication in Nature Communications.

Reviewer #2 (Remarks to the Author); expert on tumour microenvironment, obesity and cancer:

In the present study, the authors studied the crosstalk between "tumor neighboring visceral adipose stem cells" ASC and colorectal (CR) cancer stem cell (CSC) compartment. The data obtained suggest that IL6 and HGF produced by ASC expand CSC and reprogram the cells toward a mesenchymal subtype.

The present study is interesting but concerns arise concerning cell approaches and models.

Introduction section: Human adipose tissue does not contain only regulatory T cells, other lymphocyte subsets are present. It is surprising that the authors do not consider the well described accumulation of immune cells in fat depots with obesity. The term ASC must be clearly defined since the "S" actually refers to "stromal" and not to "stem" cells. Indeed, when no specific immunoselection is performed, ASC contains all human adipose tissue stroma-vascular cell subsets. To note, in the first paragraph of the results section, the term progenitor appears. The sentence "SAT and VAT react in obese state" is not clear. Obesity is an expansion of fat depot that occurs through adipocyte hypertrophy and/or hyperplasia. The term hyperplasia refers to mature adipocytes (increase in adipocyte number) and not to ASCs. Primary function of ASC is renewal of dysfunctional adipocytes and/or providing new adipocytes through adipogenesis but not paracrine effect. It is unclear what do the authors mean concerning activation of insulin like growth factor signalling in obese individuals. The term adipokines must be used with caution since it is defined as either specifically produced by mature adipocytes (mostly leptin and adiponectin) or released in the blood by adipose tissue (growth factors and cytokines exhibiting arterio-venous differences). The late definition will include IL6 but not HGF.

The study of Bhaskaran et al. published in Lancet Diabetes Endocrinol, 2018 must be considered (Association of BMI with overall and cause-specific mortality: a population-based cohort study of 3.6 million adults in the UK).

What is the origin of subcutaneous AT? Similarly, the exact anatomical origin of visceral AT is not clear and must be stated. Clinical trial number is not mentioned. In Liver, there is no evidence of

mature adipocytes but rather of liver steatosis. Please indicate the scale. Leptin immunohistochemistry is not relevant since 1) it is secreted and 2) it exhibits high non-specific signal. How is the AT coverage area performed?

“As the obese inflammatory environment is mainly sustained by the paracrine activity of adipose derived mesenchymal stem cells”: authors must add reference since inflammation in AT is thought to be sustained through adipocyte hypertrophy and immune cell accumulation.

Adipose progenitor cells are characterized as CD45-/CD34+/CD31- and not by CD105 expression since CD105 is also expressed on other cell types including monocytes and macrophages. The co-expression of CD105 and CD31 can not be considered as marker of endothelial progenitor phenotype since both are also expressed by mature endothelial cells.

Flow cytometry original dot plots must be shown. No information concerning number of viable cells seeded for culture or analysed are given. VsAT progenitor cells are described to exhibit less adipogenic capacity than the ScAT ones, what is not in agreement with the present data. The authors must discuss this point.

What are freshly purified adipose tissue cells? Please add a reference concerning expression of CD271 on ASCs. What are the levels of expression of CD271 ligands by ASCs themselves compared to CSCs? The authors mention in the text the use of CD271 antagonist but in the legend of the figure it is a neutralizing and/or blocking antibody (both are mentioned and it is unclear whether it is one or two compounds). Please provide controls showing that the antibody is indeed neutralizing/blocking CD271 activation.

Dot plot with both CD271 and VEGFR must be shown since signals appear to quite similar.

Fluorescence minus one must be shown. As shown, gating strategy is unclear.

Since ASC are not immunoselected, it is most likely that endothelial cells are present in the ASC and will proliferate. It is not demonstrated that ASC differentiate into endothelial cells.

REVIEWER COMMENTS

Reviewer #1:

After reviewing the manuscript by Di Franco S et al entitled “Adipose stem cell niche reprograms the colorectal cancer stem cell metastatic machinery” we consider it a very interesting study. It provides some new insights on CRC biology and addresses several questions that could potentially be relevant for the management of patients. However, there are several aspects that should be improved prior its publication in Nature Communications.

MAJOR CONCERNS

1. The study lacks general novelty since the contribution of obesity and adipose tissue to cancer and CRC in particular has been well documented already. Despite this fact, the present study shows some new valuable data on the cellular/molecular mechanisms ruling such influence of obesity on metastatic cancer.

We thank Reviewer #1 for the constructive comments. The novelty of our research indeed relies on the cellular/molecular mechanisms that occur in the primary CRC niche in obese patients, which lead to the reprogramming of CRC cells with expansion of the metastatic cancer cell compartment.

2. As a general idea, authors recurrently claim that cancer stem cells (CSC) are particularly relevant for the metastatic potential of CRC. It is well described that the gold standard definition of CSC is based on their tumor initiating capacity. However, authors do not show any tumor initiation assay in vivo to support their conclusions. This is fundamental to claim the role of CSC in this study. This type of assay requires the injection of tumor +/- stromal cells in a limiting dilution manner in immunodeficient mice. We recommend injection of cells in the kidney capsule to help on tumor initiation and scape from the contribution of local adipose tissue. These experiments should be done to complement several parts of the study (Figure 1F, 1G; Figure 2C; Supplementary 2; Figure 5E, 5F; Figure 6A, 6B). Only in this manner, authors will prove the effect of V-ASC, or IL6+HGF +/- Inhibitory antibodies or small drug inhibitors (Toc+Criz) on CSC biology. We know that these are very demanding experiments and therefore an alternative solution would be to eliminate the stemness concept from the whole manuscript and just focus on those capacities of tumor cells really evaluated in the study such as proliferation, invasion and metastasis.

Since our primary cancer spheres are heterogeneous and do not include exclusively cancer stem cells, we used the term “cancer stem cells” only when we referred to the CD44v6⁺ subpopulation, which we have previously shown to be endowed with a tumorigenic and metastatic potential (10.1016/j.stem.2014.01.009). Conversely, the term “ColoRectal-Cancer Sphere Cells” (CR-CSphCs) was adopted to refer to cells cultured in suspension and propagated as spheres.

However, the Reviewer #1 highlighted a very important aspect in cancer stem cell biology. The suggestion to perform subrenal capsule limiting dilution assay perfectly addresses the question: does the adipogenic niche boost the stemness of CRC cells?

Although performing *in vivo* studies during the COVID-19 pandemic was not easygoing (sorting/housing and monitoring mice), being confident that these experiments would increase the scientific impact of the manuscript, we addressed the concern raised by this reviewer accomplishing:

- a clonogenic assay using CR-CSphCs treated with V-ASC CM;
- a flow cytometry analysis of GFP expression levels in CR-CSphCs transduced with TOP-GFP lentiviral vector and treated with V-ASC CM, or control medium, to investigate the regulation of Wnt pathway activity, which we have demonstrated being a functional marker for the identification of colorectal cancer stem cells (10.1038/ncb2048; 10.1038/s41556-018-0179-z).
- *in vitro* limiting dilution assays using CR-CSphCs transduced with TOP-GFP lentiviral vector: TOPGFP^{low}; TOPGFP^{high}; TOPGFP^{low}+V-ASC CM
- *in vivo* limiting dilution assay of CR-CSphCs injected alone, or together with V-ASCs, in the subrenal capsule of immunocompromised mice.

From these experiments emerged that V-ASC CM significantly boosted the clonogenic potential of CMS2 CR-CSphCs (Figure 1A, B for reviewer). These data are included in the new Figure 1e,f of the manuscript.

Figure 1 for Reviewer. V-ASC CM boosts the clonogenic potential of CR-CSCs. A. Phase contrast analysis of CMS2 cells (CSC#9) treated with medium or V-ASC CM. Scale bars, 100 μ m. **B.** Clonogenic activity, analyzed by ELDA software, of CMS2 CR-CSCs following treatment with medium or V-ASC CM. Data are mean (Estimate) \pm standard error (lower, upper).

In vitro limiting dilution assay, showed that Wnt^{low} enriched CR-CSphCs, treated with V-ASC CM, recapitulate the clonogenic potential of Wnt^{high} cells (Figure 2 for reviewer). These data are now included in the Figure 1g of the new version of the manuscript.

Figure 2 for Reviewer. V-ASC-released cytokines reprogram the clonogenic potential of Wnt^{low} CR-CSphCs. Clonogenic assay of CMS2 CR-CSC lines (CSC#8 and #9) TOP-GFP^{high} and TOP-GFP^{low} (15% highest/lowest TOP-GFP levels) treated with medium or V-ASC CM, and evaluated by ELDA software. Data are mean (Estimate) \pm standard error (upper, lower) of 3 independent experiments.

In the Figure 1h of the new version of manuscript, we observed an increased activity of Wnt pathway in TOP-GFP transduced CMS2 CR-CSphCs treated with V-ASC CM, confirming a reprogramming of $Wnt^{inactive}$ (GFP⁻) towards Wnt^{active} (GFP⁺) cells (Figure 3 for reviewer).

Figure 3 for Reviewer. V-ASC CM drives the activation of Wnt pathway in CR-CSphCs. A. Percentage of TOP-GFP positive cells, by flow cytometry, in CMS2 cells treated with medium or V-ASC CM. **B.** Flow cytometry analysis of TOP-GFP in cells as above (black color indicates Wnt^{-} cells; green color scale indicates low, intermediate and high Wnt^{+} cells).

Of note, *in vivo* data obtained by the limiting dilution series of CR-CSphCs, alone or in combination with V-ASCs, injected into the subrenal capsule of NSG mice, showed that adipogenic niche enhances the tumorigenic and metastatic potential of CSphCs, even when a small number of cells is transplanted (Figure 1i,j of the new version of manuscript). These data indicate that CRC spheres retain cells endowed with stem cell properties, which are significantly expanded in the presence of adipose microenvironment stimuli (Figure 4 for reviewer).

Figure 4 for Reviewer. Adipogenic niche improves the tumorigenic and metastatic potential of CR-CSCs. A. Number of mouse tumor xenografts generated by subrenal capsule injection of 10, 100, 1000 or 10,000 CR-CSCs, alone or in combination with 50,000 V-ASCs. **B.** Percentage of cancer initiating cell (CIC) and its fold increase of cells as in A. Data are mean (Estimate) \pm standard error (95% confidence interval). **C.** *In vivo* imaging and immunohistochemistry analysis of xenograft tumor formation obtained by subrenal capsule injection of 100 CR-CSCs alone or together with V-ASCs at the indicated time points. Photon signal of all metastatic sites (kidney, spleen, liver, and lungs) at 12 weeks.

Please note that the figures 1e-h of the old version of the manuscript are now in the Extended Data Fig. 1j, k, m, and p of the revised version.

3. For most of the analyses performed on patient cohorts there is a serious lack of information. Authors must detail received treatments (adjuvancy?), curative surgery, and also perform a multivariate analysis including most relevant parameters that impact survival such as mutational status of KRAS, BRAF, PI3K; TNM, Grade, etc, in all studies with patient samples. This level of analysis is fundamental for concluding on the potential impact of the discoveries on CRC patients. Therefore, this should be provided for Figure 1B, Supplementary Figure 2F and Figure 6H.

To comply with the reviewer's suggestion, we have investigated the possible relevance of different clinical information, including treatment, mutational status (*BRAF*, *KRAS* or *PIK3CA*), T, N and Stage. Although the BMI was found highly significant in univariate analysis, the multivariate analysis resulted no significant whether correlated with stage, mutational background, surgery, and adjuvant therapy (see analysis below).

HR per FFMD (tot 511 cases)	Univariate analysis			Multivariate analysis		
	Patients #	HR (95% CI for HR)	p value	Patients #	HR (95% CI for HR)	p value
BMI (>=30 vs >18,5 & <30)	511	1.8 (1.3-2.4)	0.00012	174	0.403 (0.094 - 1.7)	0.22
Stage (3-4 vs 1-2)	488	2.3 (1.7-3.1)	3.10E-07		19.4 (6.3 - 59)	2.10E-07
KRAS (1 vs 0)	289	1.5 (1.1-2.2)	0.014		1.74 (0.78 - 3.9)	0.18
BRAF (1 vs 0)	250	0.65 (0.24-1.8)	0.4		1.1 (0.13 - 9.1)	0.93
PIK3CA (1 vs 0)	183	0.93 (0.33-2.7)	0.9		0.401 (0.12 - 1.4)	0.15
Curative Surgery (1 vs 0)	360	0.19 (0.1-0.36)	5.20E-07		NA	NA
Adjuvant Regimen (1 vs 0)	356	1.3 (0.95-1.8)	0.097		0.23 (0.089 - 0.59)	0.0024

Since our guidelines recommend testing for *BRAF*, *KRAS* and *NRAS* mutations only in patients with stage IV colon cancer, in the multivariate analysis the absence of correlation between BMI and tumor molecular background is likely due to the inadequate number of patients (176 out of 511) who received the molecular characterization for *KRAS*, *BRAF* and *PI3K*, which coincides with all patients in stage IV at the time of diagnosis and some other in stage I-III retrospectively analyzed to address the concern raised by this reviewer.

We next selected the variables that in the multivariate analysis reported a p value < 0.1 (Stage and Adjuvant Regimen), in addition with the curative surgery that was highly significant in univariate analysis, but cannot be determined (NA) in multivariate because all the 174 patients (for whom all the parameters were available) underwent curative surgery. By this analysis, BMI emerges as an independent negative prognostic factor for CRC patients. These data are now included in Extended Data Fig. 1a.

HR per FFMD (tot 511 cases)	Univariate analysis			Multivariate analysis		
	Patients #	HR (95% CI for HR)	p value	Patients #	HR (95% CI for HR)	p value
BMI (>=30 vs >18,5 & <30)	341	1.8 (1.3-2.6)	0.00076	341	1.82 (1.3 - 2.6)	0.00094
Stage (3-4 vs 1-2)	341	2.7 (1.9-3.7)	1.40E-08		3 (2.1 - 4.4)	8.50E-09
Curative Surgery (1 vs 0)	341	0.19 (0.097-0.36)	3.60E-07		0.109 (0.055 - 0.21)	1.80E-10
Adjuvant Regimen (1 vs 0)	341	1.3 (0.96-1.8)	0.093		0.887 (0.62 - 1.3)	0.51

Thus, in the new text we now clarify that BMI is a negative prognostic factor independent of stage and treatment.

4. Mutational profile is key for CRC progression and response to treatments. Authors should evaluate the impact of relevant oncogenic mutations for CRC on genes such as *KRAS*, *BRAF*, *PIK3CA*, *SMAD4* and *APC*, on the response to the adipocytic niche. The use of small drug inhibitors of the *KRAS/BRAF/MEK* pathway will inform on their contribution on the Adipocyte/cancer cell crosstalk.

We thank the reviewer for the very interesting comment, which could be crucial for the clinical impact of the study. To address this point and investigate the contribution of oncogenic mutations (*BRAF*, *KRAS* and *PI3K*) in the crosstalk between adipose tissue and CRC cells, we investigated whether V-ASC CM overcomes the effect of specific inhibitors such as vemurafenib, trametinib, or taselisib in

CR-CSphCs and “resistant” CR-CSphCs. Resistant cells have been isolated from liver metastasis of colon cancer patients, which did not respond to presurgical chemotherapy and bear the above-indicated mutational profile. Our data suggest that tumour adipose microenvironment protects CR-CSphCs from the targeting of the PI3K/AKT and MAPK pathways (Figure 5 for reviewer). As already reported for cytokines released by cancer-associated fibroblasts (CAFs) (see 10.1136/gutjnl-2020-323553), adipokines play a foremost role in ASC-mediated protection of CSphCs treated with specific inhibitors. Thus, tumor microenvironmental CAF or ASC cytokines convey survival signals that make CR-CSCs non-responsive to PI3K or MEK targeting. Given that CR-CSCs constitutively express high levels of HER2, which is associated with the signaling pathway activation of resistance, only targeting of HER2 in combination with PI3K and MEK inhibitors induces CR-CSC death and regression of tumour xenografts (10.1136/gutjnl-2020-323553).

Figure 5 for Reviewer. Growth factor released by ASCs protects CR-CSCs against the effect of BRAF/MEK/PI3K targeted therapies. Cell viability percentage in CR-CSCs treated with BRAFi (Vemurafenib), MEKi (Trametinib), or PI3Ki (Taselisib) in presence of V-ASC CM. Data are mean \pm SD of three independent experiments performed with *Braf*-mutant (R-CSC #4), *Kras*-mutant (R-CSC #3), *Braf/Pik3ca*-mutant (CSC #3), and *Kras/Pik3ca*-mutant (CSC #9) CR-CSCs.

5. Authors should indicate the origin of tumoroids used in all the experiments (e. g. Figure 1F-I. Are they derived from a CRC carcinoma of lean or obese patients? Are they generated from a metastasis or a primary tumor? This should be indicated for all experiments across the manuscript.

Both CMS2 and CMS4 CR-CSCs have been isolated from lean primary CRC. This information has been added in the M&M section of the manuscript.

6. Figure 1G. Quantification of proliferation in subcutaneous xenografts should be presented (e.g. Ki67 stainings) to strength the point of V-ASCs promoting tumor growth. Besides, a quantification of adipocytes should also be presented in the same xenografts. Are the injected adipocytes proliferating? Could adipocytic stroma impact on final tumor volume?

According to reviewers' suggestion, we have characterized our subcutaneous xenograft for the proliferative status. Our results, which are now included in the new version of the manuscript as Extended Data Fig. 11, show that:

- i) tumor xenografts, generated by co-injection of CR-CSphCs and V-ASCs, are endowed with a more pronounced proliferative potential as highlighted by increased number of CRC cells expressing ki67. This phenomenon was undetectable in those mouse avatars spawned by transplantation of CR-CSphCs alone or in co-injection with S-ASCs;
- ii) V-ASCs interspersed within tumor mass are characterized by a low proliferative rate, thus indicating that CRC final volume is not affected by adipose stromal tissue, which instead improves the proliferative capacity of CR-CSphCs.

The experimental procedure for quantification of CK20/ki67 positivity on paraffin-embedded sections is now reported in the M&M section.

Figure 6 for Reviewer. Visceral adipose tumor microenvironment increases the *in vivo* proliferative potential of CR-CSphCs. A. Immunohistochemical analysis of ki67 (red) and CK20 (brown) on paraffin-embedded sections of tumor xenografts generated by subcutaneous injection of CR-CSphCs alone or in combination with S-ASCs or V-ASCs. **B.** Percentage of CK20⁺/ki67⁺ cells in tumor xenografts as in A. Data are mean \pm SD of 6 independent experiments using 4 different CR-CSphC lines (CSC #1, #8, #9 and #21).

7. Figure 3F. The effect on vasculature should be quantified more precisely. In the case of tumor xenografts, a CD31 staining is recommended. Markers of ASCs should also be evaluated and quantified. It would also be important to show if the location of CD44V6+ cells is preferentially close to vascularized tumor areas (niches).

We agree with the Reviewer regarding the importance of possible co-localization of CD44v6⁺ cells and vascularized areas. Following reviewer's comment, we have performed a double staining for endothelial cells (CD31-Red) and CD44v6 CR-CSCs (CD44v6-Green).

Our data show that V-ASCs leads to formation of highly vascularized tumors, in which CD44v6⁺ cells are preferentially located at the vascular front. These data are now included in the Figure 3f of the new version of the manuscript.

The experimental procedure for quantification of vascular area has been included in the M&M section.

Figure 7 for Reviewer. The presence of ASCs within colon tumor increases the vascular surface area of neighboring CD44v6⁺ CR-CSCs. A. Immunohistochemical analysis of CD31 (red) and CD44v6 (green) on paraffin-embedded sections of tumor xenografts generated by subcutaneous injection of CR-CSCs alone or in combination with S-ASCs or V-ASCs. **B.** Percentage of vascular surface area, based on CD31 positivity, in tumor xenografts as in A. Data are mean of 6 independent experiments using 4 different CR-CSphC lines (CSC#1, #8, #9 and #21).

8. Figure 3G-I. Does IL-6+HGF treatment promote the transition from CMS2 to CMS4 signature? This gene expression profile transition is prevented by antibodies blocking IL-6 and or HGF in tumor cells exposed to V-ASC CM? This is relevant if authors want to conclude that this change in whole gene expression is ruled by IL-6 and/or HGF and required for the acquisition of metastatic potential.

In line with our functional findings, the transcriptomic analysis revealed that IL-6 and HGF drive the acquisition of EMT traits as highlighted by the concomitant downregulation of epithelial (*CDX2*, *E-CADHERIN*), and upregulation of mesenchymal (*CXCR4*, *SLUG*, *TWIST*, *ZEB1*, *ZEB2*) markers. Accordingly, double targeting of IL-6 and HGF counteracts the effects mediated by V-ASC CM in CMS2 CR-CSphCs. These data are now included in the revised version of the manuscript as Extended Data Fig. 3e.

Figure 8 for Reviewer. EMT phenotype in CMS2 CR-CSCs treated with V-ASC CM. Heatmap of epithelial (*CDX2*, *E-CADHERIN*) and mesenchymal (*CXCR4*, *SLUG*, *TWIST*, *ZEB1*, *ZEB2*) marker expression in CMS2 cells treated as indicated for 48 hours.

9. Figure 3j and Suppl. Fig.4g: Please show quantification of luciferase signal in the in vivo experiment with ZEB1 overexpressing versus control cells.

According to Reviewer's suggestion, we have now included the graph showing quantification of luciferase signal from metastatic lesions as right panel in the revised Fig. 4j and Extended Data Fig. 4i.

10. Authors conclude that obese CRC patients would be enriched in tumors with CMS4. Is this the case? Or are they CMS2 with some traits of EMT such as ZEB2 high expression? Do this potential subgroup present differential RFS or PFS?

We thank the reviewer for the comment. CMS2 CRC cells react to obese adipogenic environment by partially reprogramming their phenotype, with the acquisition of some CMS4-like EMT-related genes including ZEB-2. This point has been described in the “Discussion” section of the manuscript as follow:

“Chronic adipose-derived proteins released in TME of obese patients, reprogram CMS2 CRC cells into a cell phenotype that is characterized by the partial expression of genes associated with CMS4 signature, likely reflecting a transient CMS2/CMS4 subtype. This phenomenon is in line with the presence of CRC patients classified as mixed, whose clinical outcome remains undefined and transcriptomic profile may reside in that category named “hybrid”, “partial”, or “reversible” epithelial-mesenchymal (E/M) phenotype (10.1016/j.stem.2018.11.011; 10.1002/hep.29784).”

Our data suggest that the combinatorial exposure of CMS2 CR-CSphCs to HGF and IL-6 induce a proliferative-mesenchymal phenotype, which is consistent with the so-called “hybrid-EMT”. This observation is highly relevant from a clinical point of view, and it will be the subject of a future research project. The complexity in determining this phenotype in our model arise from the current unavailability of a database containing survival information (PFS or RFS) + RNAseq data + BMI information. We predict that the future availability of this information will allow to define a new signature specific for obese CRC patients, and to determine if CMS2 CRC patients progress differently when belonging to lean versus obese patients.

MINOR

1. Figure 1C. A magnification of CDX2 staining in the metastatic tumors will be appreciated. The current image is not clear enough.

We have now added a magnification of CDX2 staining.

2. Figure 1D. The Leptin staining looks with high unspecific background. It would be much appreciated to present the results with another antibody marking adipocytes or an alternative technique (e.g., RT-PCR).

We thank the reviewer for the relevant comment, following which we decided to stain our primary and metastatic colon cancer tissues with Adiponectin antibody to identify adipose cells. The Figure 1D was substituted with the triple staining for CD34/CD31/CD45 (see answer to point n.10 of referee#2) and the adiponectin staining revealing the presence of adipose cells, within colon cancer cells, was moved in Supplementary Figure 1C.

3. Figure 2. The conclusions on the CD44V6+ population would be very much strengthened by showing their increased presence in CRC carcinomas of obese patients.

We have retrospectively analyzed our tissue biobank of lean and obese CRC patients and performed an immunohistochemistry analysis of CD44v6 expression. Imaging analysis has been scored as quick score (Q), by multiplying the percentage of positive cells (P) by the intensity (I). Analysis of a cohort of 37 tissue samples displayed a positive correlation between CD44v6 expression and obese CRC patients. These data have been included in Extended Data Fig. 4d of the new version of the manuscript.

Figure 9 for Reviewer. CD44v6 expression in obese CRC patients. A. Immunohistochemical analysis of CD44v6 on CRC tissue with weak (1), moderate (2), and strong (3) staining intensity. Scale bars represent 100 μm. **B.** CD44v6 score (Q score = percentage of positive cells (P) x intensity (I)) in lean (n=28) and obese (n=9) CRC patients.

4. Supplementary Figure 2a. The staining of nuclear beta-catenin is not convincing. Are these cells APC mutant? This should be quantified by western blot or other means. Nuclear beta-catenin accumulation is not a definitive marker of “stemness”. PDOs from #8 should be also characterized for beta-catenin.

We thank you for the comment. As highlighted by this Reviewer, the immunohistochemical analysis of nuclear β -catenin, although showing decreases levels in differentiated cells, does not give any information about quantification of nuclear expression. In accordance with our previous evidence (10.1038/ncb2048; 10.1038/s41556-018-0179-z), western blot analysis revealed that CR-CSphCs express high levels of active (non-phosphorylated) β -catenin compared to SDACs, which are endowed with a non-active (phosphorylated) isoform, indicating that β -catenin/Wnt pathway is a functional marker of CR-CSCs (new Extended Data Fig. 2a).

Figure 10 for Reviewer. *In vitro* CR-CSC differentiation setting. A. Immunoblot analysis of non phospho-active and total β -catenin in CMS2 CR-CSCs (CSC #8 and #9) and their sphere-derived adherent cells (SDACs). β -actin was used as loading control. (lower panel) Phase contrast analysis of cells as above. Scale bars, 200 μ m.

5. Figure 3d. Images are not informative and could be removed or substituted by a flow cytometry analysis of CD31. There is a label mistake on the quantification plot showing the cells treated with anti-VEGF antibody.

To address this comment and to exclude the possibility that the increase of CD31⁺ cells is driven by an increase of pre-existing CD31⁺ cells, we have depleted this cell population (new Extended Data Fig. 3c) and following exposure to conditioned medium (CM) derived from CD44v6⁺ cells, we observed a change of enriched CD34⁺/CD31⁻ ASCs into CD31⁺ endothelial-like cells (new Figure 3d; Figure 11 for Reviewer).

Figure 11 for Reviewer. CD44v6⁺ CR-CSC released VEGF induces the expression of CD31 in CD34⁺/CD31⁻/CD45⁻ ASCs. Percentage of CD31 positivity, by flow cytometry analysis, on CD34⁺/CD31⁻/CD45⁻ enriched ASCs exposed to vehicle (Medium), CD44v6⁺ CR-CSCs CM, in presence or absence of VEGF neutralizing antibody, or VEGF for 14 days. Data are mean \pm SD of three independent experiments using CR-CSCs CM derived from four different cell lines (#1, #8, #9, #21).

6. Figure 4a: Gene sets positively enriched by V-ASC-CM do not clearly show a relationship with EMT. We recommend to move Supplementary Figure 3d in to the main Figure 4 to better support this conclusion.

We thank the reviewer for the comment. We replaced Figure 4a with the new Extended Data Fig. 3f, to increase the clarity of manuscript conclusions.

7. Figure 3b: Gene names are shown in a letter size too small. They cannot be read. Please magnify.

According to Reviewer comment, we increased the size of Figure 3a as much as possible to fit into the Figure, without altering the setting of the panel.

8. Figure 4e: Please quantify changes in ZEB1 and ZEB2 protein expression.

We thank the Reviewer for the comment. We have added a panel in Figure 4e showing the number of CR-CSphCs expressing nuclear ZEB1 or ZEB2. Quantification of ZEB2 protein expression is reported in Figure 4h.

Reviewer #2:

In the present study, the authors studied the crosstalk between “tumor neighboring visceral adipose stem cells” ASC and colorectal (CR) cancer stem cell (CSC) compartment. The data obtained suggest that IL6 and HGF produced by ASC expand CSC and reprogram the cells toward a mesenchymal subtype.

1. The present study is interesting, but concerns arise concerning cell approaches and models. Introduction section: Human adipose tissue does not contain only regulatory T cells, other lymphocyte subsets are present. It is surprising that the authors do not consider the well described accumulation of immune cells in fat depots with obesity.

We have adjusted the text to not exclude any T cell compartments, as follow:

“White AT is characterized by a marked cell variety, which includes adipocytes, T cells, endothelial cells, fibroblasts, macrophages and adipose stromal cells (ASCs) (10.1016/j.trecan.2018.03.004).”

2. The term ASC must be clearly defined since the “S” actually refers to “stromal” and not to “stem” cells. Indeed, when no specific immunoselection is performed, ASC contains all human adipose tissue stroma-vascular cell subsets. To note, in the first paragraph of the results section, the term progenitor appears.

The reviewer is correct in this point. Given the absence of an immunoselection protocol to specifically sort the Adipose Stem Cells (CD90⁺, CD105⁺, CD73⁺, CD271⁺, CD10⁺, CD200⁺, CD34⁺, CD36⁺, CD106⁻, CD45⁻, CD31⁻; 10.1080/14653240600855905; 10.1038/s41467-019-09992-3), the acronym “ASC” in the revised manuscript stands for Adipose Stromal Cells. To demonstrate that our cell lines are enriched in a cell subset expressing the above-mentioned adipose stem cell markers, we performed a new ASC characterization that is reported in the new Extended Data Fig. 1f.

Figure 1 for Reviewer. Mesenchymal and endothelial markers expression in S- and V-ASCs. Flow cytometry analysis of the indicated mesenchymal stem cell and endothelial putative markers in ASCs isolated from subcutaneous (S-ASCs; n= 13) and visceral (V-ASCs; n= 10) adipose tissue.

- 3. The sentence “SAT and VAT react in obese state” is not clear. Obesity is an expansion of fat depot that occurs through adipocyte hypertrophy and/or hyperplasia. The term hyperplasia refers to mature adipocytes (increase in adipocyte number) and not to ASCs. Primary function of ASC is renewal of dysfunctional adipocytes and/or providing new adipocytes through adipogenesis but not paracrine effect.**

We thank the reviewer to give us the chance to clarify this point and to make it clearer to the reader. We adjusted the text based on what Jeffery and colleagues (10.1038/ncb3122) recently reported about the different reaction of subcutaneous and visceral adipose tissue to overnutrition. While both subcutaneous and visceral white adipose tissues expand by hypertrophy of pre-existing adipocytes, only visceral fat can determine a concomitant hyperplastic response, which is driven by adipose precursor cells, identified as Lin⁻ Sca1⁺ CD29⁺ CD34⁺. This phenomenon could be due to the different embryological origin of subcutaneous and visceral adipose tissue, or by the presence of resident factors. In particular, mature adipocytes are post-mitotic cells, thus suggesting that hyperplasia arises from the expansion and differentiation of adipocyte precursors (10.1016/j.cell.2008.09.036; 10.1038/nrm3198). Mature adipocytes together with ASCs influence the surrounding cell populations through the release of a plethora of inflammatory and angiogenic cytokines (10.1210/er.2010-0030).

- 4. It is unclear what do the authors mean concerning activation of insulin like growth factor signaling in obese individuals.**

Given the controversial role of IGF and its related pathway in obesity, which is out of the scope of the manuscript, we revised the sentence in the manuscript, to avoid any confusion in the interpretation, in the following:

“In obese individuals, adipose-released proteins, including TNF- α , IL-6 and monocyte chemo-attractant protein1 (MCP1), promote a chronic inflammatory state that creates a microenvironment able to sustain tumor progression (10.1038/s41575-018-0069-7; 10.1038/s41574-018-0126-x”

- 5. The term adipokines must be used with caution since it is defined as either specifically produced by mature adipocytes (mostly leptin and adiponectin) or released in the blood by adipose tissue (growth factors and cytokines exhibiting arterio-venous differences). The late definition will include IL6 but not HGF.**

We agree with this reviewer about the disputable definition of the term “adipokine”. Although HGF is commonly described as adipokine, when it is released by adipose cells (10.1158/0008-5472.CAN-11-2771; 10.3389/fendo.2018.00503) instead of a growth factor. We replaced the term “adipokine” with “proteins/factors released by....(i.e. IL-6 and HGF)” to make the manuscript clearer to reader.

- 6. The study of Bhaskaran et al. published in Lancet Diabetes Endocrinol, 2018 must be considered (Association of BMI with overall and cause-specific mortality: a population-based cohort study of 3.6 million adults in the UK).**

We have now included and discussed the above-mentioned reference in the Introduction section as follow:

“Indeed, several retrospective studies of a large cohort of cancer patients highlighted that obesity has a significant impact on overall survival, posing this indicator as a significant negative

prognostic factor, including for CRC patients (10.1093/aje/152.9.847; 10.1200/JCO.2011.38.0287; 10.1016/S2213-8587(18)30288-2). Furthermore, Bhaskaran and colleagues showed that while in CRC patients with BMI from 15 to 25 kg/m², the mortality risk does not vary, it linearly increases in those with BMI from 25 to 50 kg/m² (10.1016/S2213-8587(18)30288-2)”

7. What is the origin of subcutaneous AT? Similarly, the exact anatomical origin of visceral AT is not clear and must be stated.

Information about the origin of subcutaneous and visceral adipose tissue has been now added to the M&M section of the manuscript. Please find here the description as reference:

“Adipose stromal cells from VAT and SAT have been obtained from greater omentum and subcutaneous anterior abdominal wall, respectively.”

8. In Liver, there is no evidence of mature adipocytes but rather of liver steatosis. Please indicate the scale. Leptin immunohistochemistry is not relevant since 1) it is secreted and 2) it exhibits high non-specific signal. How is the AT coverage area performed?

In Extended Data Fig. 1c, we replaced the Leptin with adiponectin staining. The immunohistochemical analysis using Adiponectin antibody clearly show better results, with a well-defined staining and the absence of background signal. These experimental details have been now included in the M&M section as follow: “The adipose tissue coverage area has been evaluated by calculating the area covered by Adiponectin⁺ (ImageJ, Colour deconvolution plugin).”

Accordingly, Extended Data Fig. 1d legend has been modified as follow: “Percentage of adipose tissue (AT) area based on adiponectin positivity, evaluated in primary tumor and liver metastasis paraffin-embedded sections from patients as in (c).”. The experimental procedure for quantification of adipose coverage area has been included in the M&M section.

Figure 2 for Reviewer. Adipose cells are enriched in primary and metastatic obese CRC. A. Immunohistochemical analysis of Adiponectin and CDX2 on primary and liver metastasis tissue specimens from obese CRC patients. **B.** Percentage of adipose tissue (AT) area based on adiponectin positivity, evaluated in paraffin-embedded sections of specimens as in A.

9. **“As the obese inflammatory environment is mainly sustained by the paracrine activity of adipose derived mesenchymal stem cells”:** authors must add reference since inflammation in AT is thought to be sustained through adipocyte hypertrophy and immune cell accumulation.

We adjusted the text according to the reviewer’s comment as follow:

“As the obese inflammatory environment is strongly sustained by the paracrine activity of either mature adipocytes and adipose-derived vascular stromal cells³⁴, including adipose stem cells, we next investigated whether this cell subset is present in the tumor area and could influence cancer cell phenotype.”

10. **Adipose progenitor cells are characterized as CD45-/CD34+/CD31- and not by CD105 expression since CD105 is also expressed on other cell types including monocytes and macrophages.**

We agree with this reviewer and as already reported by Estève et al. 2019 (10.1038/s41467-019-09992-3), we added a triple immunostaining for CD45/CD34/CD31. Cells expressing CD34⁺/CD31⁻/CD45⁻ indicate the presence of adipose progenitor cells interspersed within the primary and metastatic lesions. These data are now included in Fig. 1d and Extended Data Fig. 1e of the revised version of the manuscript. The experimental procedure for quantification of CD34/CD31/CD45 positivity on paraffin-embedded sections is now reported in the M&M section.

Figure 3 for Reviewer. Obese CRC patients show increase number of adipose progenitor cells. Immunohistochemical analysis and number of CD34 (brown color), CD31 (green color) and CD45 (red color) positive cells on primary and liver metastasis CRC paraffin-embedded sections from lean and obese patients. Data are mean \pm S.D. of independent experiments using tissue specimens from 9 CRC patients. Scale bars, 100 μ m.

11. **The co-expression of CD105 and CD31 cannot be considered as marker of endothelial progenitor phenotype since both are also expressed by mature endothelial cells.**

According to Reviewer’s comment, we modified the text by removing the term “progenitor”, as follows:

“Conversely, tumor specimens of lean CRC patients displayed the presence of cells expressing CD34+/CD31+/CD45- ascribable to an endothelial phenotype (Fig. 1d and Extended Data Fig. 1e).”
(See Figure 3 for reviewer above).

12. Flow cytometry original dot plots must be shown.

Original dot plots of the flow cytometry analysis (Figure 4i and Extended Data Fig. 4c, Figure 3c, Figure 3d and Extended Data Fig. 3c) have been included in the revised version of the manuscript.

13. No information concerning number of viable cells seeded for culture or analyzed are given.

The information regarding the number of viable cells seeded for culture and analyses is included in the M&M section of the revised manuscript as follows:

“Viability of CR-CSphC culture, which is routinely estimated by trypan blue and 7-AAD, is 93±7 %.”

14. VsAT progenitor cells are described to exhibit less adipogenic capacity than the ScAT ones, what is not in agreement with the present data. The authors must discuss this point.

The adipogenic potential of subcutaneous and visceral adipose-derived mesenchymal stem cells is widely debated, with some research groups showing that subcutaneous mesenchymal cells have more pronounced adipogenic potential (10.1371/journal.pone.0036569), and other showing that visceral mesenchymal stem cells are endowed with higher differentiative potential into adipocytes (10.3390/cells8101288).

Our previous experimental setting showed that, up to 21 days, V-ASC progenitor cells retained an earlier adipogenic potential compared with those derived from S-ASCs, which exhibited high number of adipose cells, characterized by small lipid droplets, thus indicating the presence of immature adipocytes. However, in agreement with the reviewer concern, the prolonged exposure to adipocyte differentiation medium highlighted that S-ASC progenitors, at 28 days, are endowed with equal

adipocyte differentiation capacity to that of V-ASC progenitor cells. The revised manuscript now includes the new, following panels as Extended Data Figure 1h,i:

Figure 4 for Reviewer. S- and V-ASC are endowed with comparable adipogenic differentiation potential. A. Percentage of adipocyte differentiation potential of S-ASCs or V-ASCs at 21 days in the old and at 28 days in the new Supplementary Figure 1F, G. Data are mean \pm S.D. of 4 independent experiments using 3 different S-ASC and V-ASC cultures. **B.** Phase contrast and fluorescence analysis of lipid droplets content, detected by AdipoRed staining, in cells as indicated, showing the increase of lipid droplets content in the S-ASCs at 28 days. Nuclei were counterstained with DAPI. Scale bars, 100 μ m.

15. What are freshly purified adipose tissue cells?

Adipose stromal cells have been isolated from AT and used in their first passages, as indicated in M&M section (passage 1-12). To make it clearer, we changed the “freshly purified adipose tissue cells” with “adipose stromal cells (ASCs)” along the manuscript.

16. Please add a reference concerning expression of CD271 on ASCs.

Although the established heterogeneous expression of CD271 in adipose stromal cells, we added references of CD271 expression in ASCs, as suggested by this reviewer (10.1038/s41598-018-27587-8; 10.1038/s41467-019-09992-3; 10.1002/jcb.26496).

17. What are the levels of expression of CD271 ligands by ASCs themselves compared to CSCs?

Following the reviewer suggestion, in the new Figure 21 we showed that levels of CD271 ligands released by ASCs are negligible if compared with those derived from CR-CSCs (see Figure 4 for reviewer).

Figure 5 for Reviewer. Release of neurotrophins (NGF, BDNF, NT-3, and NT-4) in CR-CSCs is boosted by V-ASC CM. Lollipop plot showing the amount of NGF, BDNF, NT-3 and NT-4 released by the indicated cells, in presence or absence of V-ASC CM. Data are mean of 3 independent experiments using CR-CSPhCs from different patients (#1, #8, #9, #21).

18. The authors mention in the text the use of CD271 antagonist but in the legend of the figure it is a neutralizing and/or blocking antibody (both are mentioned, and it is unclear whether it is one or two compounds). Please provide controls showing that the antibody is indeed neutralizing/blocking CD271 activation.

We are sorry for the confusion about the terms used along the manuscript to describe the neutralizing antibody against CD271. The antibody used to neutralize the activation of CD271 (ME20.4, mouse IgG1, Merck) has been published and validated for its neutralizing activity against the neurotrophin receptor (10.1073/pnas.81.21.6681, Fig. 2a, b). However, to further validate its neutralizing activity, we investigated the proliferative rate of ASCs upon exposure to exogenous NGF in presence or absence of α -CD271 (0.5 μ g/ml). We observed that neutralization of CD271 prevented the enhanced proliferation sustained by exogenous NGF of both VsAT and ScAT progenitors.

Figure 6 for Reviewer. Neutralizing activity of α -CD271. Cell growth of S- and V-ASCs treated with medium or NGF in presence of α -CD271 neutralizing antibody at the indicated time points. Data are mean \pm standard deviation of 3 independent experiments performed with 5 different S- and V-ASC lines.

19. Dot plot with both CD271 and VEGFR must be shown since signals appear to quite similar. Fluorescence minus one must be shown. As shown, gating strategy is unclear.

Figure 3c now includes the Fluorescence Minus One (FMO) analysis and the gating strategy of CD271 and VEGFR expression on ASCs.

Figure 7 for Reviewer. The majority of CD271⁺ ASCs express VEGFR. Gating strategy of CD271/VEGFR expression on ASCs. (*middle panels*) Dot-plots of CD271/VEGFR staining with or without the indicated antibody (FMO-APC control, minus CD271-PE-Cy7). (*lower panels*) Flow cytometry analysis of VEGFR in gated CD271⁺ ASCs. Numbers of positive cells indicate that the majority of CD271⁺ cells co-express VEGFR.

20. Since ASC are not immunoselected, it is most likely that endothelial cells are present in the ASC and will proliferate. It is not demonstrated that ASC differentiate into endothelial cells.

As also suggested by reviewer #1, to address this comment and to exclude the possibility that the increase of CD31⁺ cells is driven by an increase of pre-existing CD31⁺ cells, we have depleted this cell population (new Extended Data Fig. 3c). Following exposure to CD44v6⁺ cells CM, we observed a change of enriched CD34⁺/CD31⁻ ASCs into CD31⁺ endothelial-like cells (new Figure 3d; Figure 8 for Reviewer).

Figure 8 for Reviewer. CD44v6⁺ CR-CSC released VEGF induces the expression of CD31 in CD34⁺/CD31⁻/CD45⁻ ASCs. Percentage of CD31 positivity, by flow cytometry analysis, on CD34⁺/CD31⁻/CD45⁻ enriched ASCs exposed to vehicle (Medium), CD44v6⁺ CR-CSCs CM, in presence or absence of VEGF neutralizing antibody, or VEGF for 14 days. Data are mean ± SD of three independent experiments using CR-CSCs CM derived from four different cell lines (#1, #8, #9, #21).

REVIEWER COMMENTS

Reviewer #1 (Remarks to the Author):

After a close review of the new manuscript version, it is evident that authors have answered most of the initial concerns. The amount and quality of the new provided data is excellent. I am convinced that now the manuscript is ready for being published in Nature Communications.

Héctor G. Palmer, PhD.

Reviewer #2 (Remarks to the Author):

The authors provided a detailed response and added new data as requested that improved the revised version of the manuscript. Some minor concerns are still present in the new version:

« White AT is characterized by a marked cell variety, which includes adipocytes, T cells, endothelial cells, fibroblasts, macrophages, and adipose adipose stromal cells». Please change for example to adipocytes, immune cells, vascular and progenitor cells . Indeed for lymphocytes, not only T lymphocytes are present but B lymphocytes as well as NK cells for example, fibroblasts are not completely defined cell subtype and thought to be part of progenitors, finally the term « adipose stromal cells » includes all non-adipocyte stroma-vascular cells obtained after digestion.

« Visceral fat is also able to determine a hyperplastic response, which is driven by adipose precursor cells, identified as Lin⁻ Sca1⁺ CD29⁺ CD34⁺» It must be clearly stated that this sentence refers to mouse model of diet-induced obesity. Same remark for the following sentence « This phenomenon could be due to the different embryological origin of subcutaneous and visceral adipose tissue, AND/or by the presence of specific resident factors ». Overall, It will help to clearly state the species/models.

REVIEWER COMMENTS

Reviewer #1 (Remarks to the Author):

After a close review of the new manuscript version, it is evident that authors have answered most of the initial concerns. The amount and quality of the new provided data is excellent.

I am convinced that now the manuscript is ready for being published in Nature Communications

Héctor G. Palmer, PhD.

We thank the Reviewer #1 for the positive comments and the careful review, which improved the clarity and scientific robustness of the manuscript.

Reviewer #2 (Remarks to the Author):

The authors provided a detailed response and added new data as requested that improved the revised version of the manuscript. Some minor concerns are still present in the new version:

« White AT is characterized by a marked cell variety, which includes adipocytes, T cells, endothelial cells, fibroblasts, macrophages, and adipose adipose stromal cells».Please change for example to adipocytes, immune cells, vascular and progenitor cells . Indeed for lymphocytes, not only T lymphocytes are present but B lymphocytes as well as NK cells for example, fibroblasts are not completely defined cell subtype and thought to be part of progenitors, finally the term « adipose stromal cells » includes all non-adipocyte stroma-vascular cells obtained after digestion.

« Visceral fat is also able to determine a hyperplastic response, which is driven by adipose precursor cells, identified as Lin⁻ Sca1⁺ CD29⁺ CD34⁺» It must be clearly stated that this sentence refers to mouse model of diet-induced obesity. Same remark for the following sentence « This phenomenon could be due to the different embryological origin of subcutaneous and visceral adipose tissue, AND/or by the presence of specific resident factors ». Overall, It will help to clearly state the species/models.

We thank the Reviewer #2 for the focused and constructive comments. As suggested, we adjusted the text as follow:

- White AT is characterized by a marked cell variety, which includes adipocytes, **immune cells**, vascular and progenitor cells (Lengyel et al., 2018).
- **Recent evidence showed that** visceral fat, in **high-fat diet induced obese mouse models**, determines a hyperplastic response, which is driven by adipose precursor cells, identified as Lin⁻ Sca1⁺ CD29⁺ CD34⁺ (Jeffery et al., 2015).
- This phenomenon could be due to the different embryological origin of subcutaneous and visceral adipose tissue, **and/or** by the presence of specific resident factors, **as highlighted by lineage tracing experiment performed in adult Wt1-GFP knock-in mice (Chau et al., 2014).**